# LightTransfer: Your Long-Context LLM is Secretly a Hybrid Model with Effortless Adaptation

**Xuan Zhang***                                    *xuanzhang.2020@phdcs.smu.edu.sg*
*Singapore Management University*

**Fengzhuo Zhang**                                           *fzzhang@u.nus.edu*
*National University of Singapore*

**Cunxiao Du** [†]                                              *ducx@sea.com*
*Sea AI Lab, Singapore*

**Chao Du**                                                   *duchao@sea.com*
*Sea AI Lab, Singapore*

**Tianyu Pang**                                          *tianyupang@sea.com*
*Sea AI Lab, Singapore*

**Wei Gao**                                               *weigao@smu.edu.sg*
*Singapore Management University*

**Min Lin**                                                  *linmin@sea.com*
*Sea AI Lab, Singapore*

**Reviewed on OpenReview:** *https://openreview.net/forum?id=kne4vWICrO*

## Abstract

Scaling language models to handle longer contexts introduces substantial memory challenges due to the growing cost of key-value (KV) caches. Motivated by the efficiency gains of hybrid models and the broad availability of pretrained large transformer backbones, we explore transitioning transformer models into hybrid architectures for a more efficient generation. In this work, we propose LIGHTTRANSFER, a lightweight method that transforms models such as LLaMA into hybrid variants. Our approach identifies *lazy* layers—those focusing on recent or initial tokens—and replaces their full attention with streaming attention. This transformation can be performed without any training for long-context understanding tasks or with minimal fine-tuning for o1-like long reasoning generation tasks that require stronger reasoning capabilities. Experiments across diverse benchmarks and models (e.g., LLaMA, Mistral, QwQ-STILL) demonstrate that, even when half of the layers are identified as *lazy*, LIGHTTRANSFER achieves up to 2.17× throughput improvement with minimal performance loss ($< 1.5\%$ on LongBench) and achieves 53.3% on math benchmark AIME24 of advanced o1-like long reasoning model QwQ-STILL. Our project homepage: `https://sites.google.com/view/lighttransfer`.

## 1 Introduction

Recent advancements in large language models (LLMs) have extended their capacity for handling long context inputs and generating long-form reasoning. For example, LLaMA-3.1 supports context lengths up

---

*Work done during Xuan Zhang's associate membership at Sea AI Lab.

[†]Correspondence to Cunxiao Du.

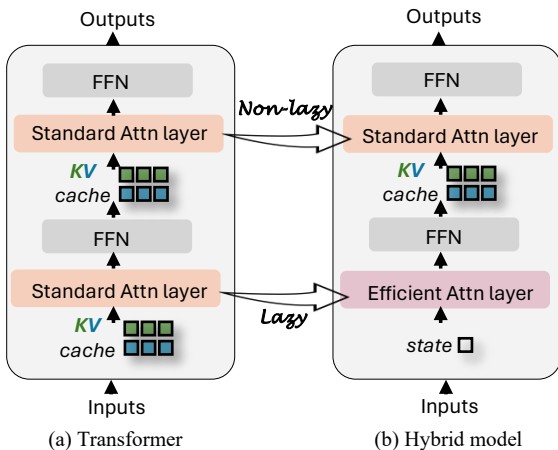

Figure 1: (a) A standard transformer architecture. (b) A hybrid model in which certain layers of a standard transformer are replaced with more memory-efficient designs. LIGHTTRANSFER identifies lazy layers in (a) and transforms them into more efficient variants, yielding (b).

to 128K (Dubey et al., 2024), while OpenAI's o1 can produce sequences of up to 100K tokens (OpenAI, 2024). As the cornerstone of the efficient inference of these models on long context, key-value (KV) cache stores precomputed key and value tensors for each token in the language sequence to avoid recomputing them for each attention layer. However, as the number of model layers and input lengths increases, the memory required for storing the KV cache grows significantly, posing challenges for inference efficiency.

Various methods have been proposed to reduce the KV cache storage by modifying the model architecture (Shazeer, 2019; Brandon et al., 2024; Goldstein et al., 2024; Nawrot et al., 2024; Wang et al., 2024b; Yu et al., 2024). One promising approach is the hybrid model. As shown in Figure 1, in these hybrid models, certain layers of a standard transformer are replaced with more memory-efficient mechanisms such as RNNs (Sherstinsky, 2020), Mamba (Gu & Dao, 2023), and sliding window attention (Beltagy et al., 2020). These approaches exploit the notion that different layers can be manually assigned distinct functionalities, such as using memory-efficient mechanisms for local context processing and standard attention for global context handling (Gemma et al., 2024), thereby achieving notable memory savings. Concrete examples of such hybrid architectures include Transformer-Recurrent Neural Network (RNN) designs such as YoCo (Sun et al., 2024), Transformer-Mamba approaches such as Jamba (Lieber et al., 2024; Team et al., 2024), and Transformer-Sliding Window models like Gemma 2 (Gemma et al., 2024) and YoCo (Sun et al., 2024). However, a key limitation is that they require training the entire model from scratch.

Given the substantial efficiency gains offered by hybrid models and the availability of large-scale pretrained transformer backbones, a natural direction is to transition these pretrained models into hybrid architectures with minimal additional training. A straightforward method is to replace traditional full attention layers with sparse attention, thereby adopting a fixed-size KV cache to reduce memory overhead. A representative example is *streaming attention* (Xiao et al., 2023), which augments the sliding-window mechanism by introducing *sink tokens*. However, as Table 2 shows, completely substituting all standard attention layers with streaming attention leads to a severe degradation in the model's ability to process long contexts, thereby undermining its capacity to capture global information. Consequently, when transitioning from a pretrained transformer to a hybrid model, two primary challenges arise. First, it is necessary to retain some standard attention layers to preserve the model's long-context modeling capabilities, raising the critical question: which layers should remain intact? Second, this transition should ideally be lightweight, enabling efficient adaptation with minimal data or even allowing it to be applied entirely at test time. Otherwise, if large-scale pretraining data were required, one could simply train a hybrid model from scratch, undermining the value of a transition-based approach.

In response to the above challenges, we examine the attention patterns in different transformer layers to determine whether each layer exhibits distinct functionalities. We conduct preliminary experiments (Section 4) and identified two key findings: First, *certain layers in long-context LLMs exhibit lazy behavior*, primarily focusing on semantically unimportant tokens (e.g., the initial few tokens) and the most recent during answer generation. The properties of lazy layers address our first challenge and enable standard transformer-based

LLMs to operate in a hybrid-like manner: identified lazy layers use streaming attention, whereas non-lazy layers retain full attention. Second, after analyzing attention weight patterns, we find that *layer behavior is consistent across tokens for a given long input.* This insight partially addresses the second challenge and paves the way for a test-time transformation in which selective modifications are applied during the prefilling stage, allowing for efficient adaptation without extensive retraining.

Building upon this insight, we propose LIGHTTRANSFER, a lightweight method that transforms models such as *LLaMA*, *Mistral* (Jiang et al., 2023), and *QwQ* (Qwen, 2024b) into their corresponding hybrid variants. Specifically, as shown in Figure 1, we analyze the attention allocation patterns in each layer to determine whether it can be treated as a lazy layer. In lazy layers, we apply streaming attention, while standard attention is retained in non-lazy layers. The output of the transformer with a reduced KV cache differs from the original output due to the reduced cache size, and this difference is theoretically analyzed in Theorem 5.1. For tasks where the input is sufficiently long (i.e., long-context understanding), we leverage on-the-fly lazy layer identification at the prefilling stage, LIGHTTRANSFER-TEST. In addition, for o1-like long reasoning generation tasks, even though the questions can be relatively short (only a few dozen tokens) yet demand higher model capacity, we surprisingly find that minimal training still enables robust performance (LIGHTTRANSFER-TRAIN). In practice, this transition requires only around 5K samples (originally utilized for long-reasoning ability distillation (Min et al., 2024)), underscoring the lightweight nature of our approach.

We conduct experiments on four representative LLMs (i.e., LLaMA2-7B-chat (Touvron et al., 2023), Mistral-7B-Instruct (Jiang et al., 2023), LLaMA3-8B-Instruct and its 70B counterpart (Dubey et al., 2024)), evaluating them on long-context benchmarks including LongBench (Bai et al., 2023) and Needle-In-A-Haystack (NIAH) (Kamradt, 2023). In addition, we adapt an o1-like long reasoning model QwQ-32B-STILL and assess its performance on MATH-OAI (Lightman et al.), AIME24 [1], and GSM8K (Cobbe et al., 2021). Experimental results indicate that hybrid models converted via LIGHTTRANSFER achieve performance on par with their standard transformer counterparts. For example, on long-context understanding tasks, it achieves only a 1.45% performance decline on LongBench. For long reasoning tasks, it achieves performance that is comparable to or even better on the widely used mathematical benchmark AIME24, reaching 53.3% accuracy. Notably, these results were obtained while half of the model's layers employed streaming attention, yielding up to a 2.17× increase in throughput.

## 2   Related Works

Memory-efficient architectures, such as linear RNN-based architectures (e.g., Mamba (Gu & Dao, 2023)) and those employing sparse attention methods (e.g., streaming attention (Xiao et al., 2023)), have demonstrated clear advantages in deployment, including reduced memory usage and higher throughput (Peng et al., 2023; Dao & Gu, 2024; Yang et al., 2023; Sun et al., 2024; Lieber et al., 2024; Gemma et al., 2024). However, a key drawback of these memory-efficient models is their limited ability to handle extended contexts effectively (Behrouz et al., 2024; Yuan et al., 2024). Meanwhile, the inference memory cost of standard transformers (i.e., the storage of the KV cache) grows linearly as the context length increases (Shi et al., 2024; Li et al., 2024c;a). To address these challenges, recent research has proposed hybrid architecture: maintaining the strong capabilities of pretrained transformers while selectively substituting certain transformer layers with more memory-efficient modules, thereby balancing high performance with practical deployment efficiency (Lieber et al., 2024; Gemma et al., 2024; Sun et al., 2024; Botev et al., 2024; De et al., 2024). For example, Jamba (Lieber et al., 2024; Team et al., 2024) integrates Mamba (Gu & Dao, 2023) with transformer, Gemma 2 (Gemma et al., 2024) alternates sliding-window attention with standard attention layers, and Minimax-01 (Li et al., 2025) employs lightning attention (Qin et al., 2024) in certain layers. However, a key limitation of these methods is their reliance on training the entire model from scratch. Although recent approaches aim to leverage the capabilities of large-scale pretrained models by converting selected layers into memory-efficient structures to form a well-trained hybrid model, they still depend on extensive training data (Wang et al., 2024a; Ge et al., 2024). For instance, LongGen (Ge et al., 2024) transforms certain layers in pretrained LLM into sparse attention but requires retraining on over 2TB of data. Differently, our LIGHTTRANSFER framework is designed to be substantially more lightweight. Despite requiring no additional

---

[1] https://huggingface.co/datasets/AI-MO/aimo-validation-amc

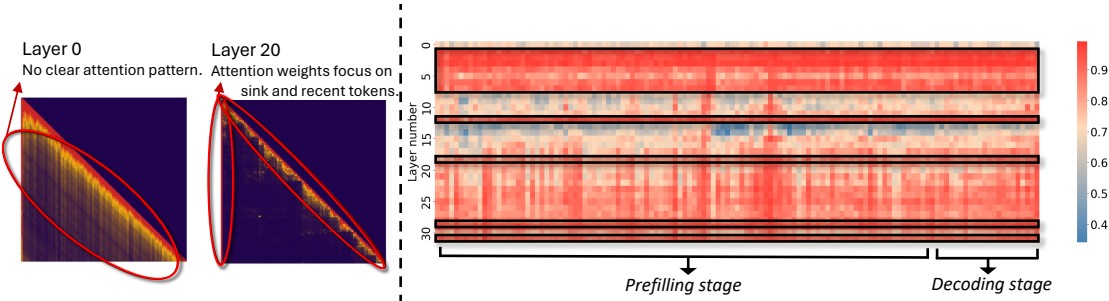

Figure 2: Visualization of attention weight distributions on LLaMA3-8B. Left: The attention patterns across different layers. Right: Each cell in the heatmap represents the aggregated attention weight from each query token (x-axis) to two token groups: (1) the initial $W_{sink}$ tokens, and (2) the most recent $W_{recent}$ tokens, during both the prefilling and decoding stages. Layers predominantly attending to these token groups are highlighted with black outlines. The color intensity indicates the sum of attention weights, averaged across all samples in a batch.

training for long-context understanding tasks, and only 5K training examples for more demanding long-text reasoning tasks (as originally used for long-reasoning ability distillation (Min et al., 2024)), it still achieves strong performance on both fronts. The superiority of our LIGHTTRANSFER comes from the identification of each layer's function, whereas LongGen always uses a fixed structure (retaining the middle layers for full attention). Some works also attempt to fully transfer transformer models into RNN-like architectures (Kasai et al., 2021; Zhang et al., 2024b; Mercat et al., 2024; Zhang et al., 2024a; Bick et al., 2024). However, these methods primarily focus on short-context tasks (e.g., QA), whereas our approach targets long-context scenarios.

## 3 Preliminary

Before introducing LIGHTTRANSFER, we provide a brief overview of the generative inference in autoregressive LLMs, which is the key background for our method.

**Inference stages.** The typical generative LLM inference process involves two stages: (1) *Prefilling*: the autoregressive LLM processes the input prompt $X$ by parallel computing, and also saves the KV cache of tokens in $X$. The output of the last token in this stage is the first token of the response. (2) *Decoding*: after the prefilling stage is completed, the LLM generates output tokens one by one, and saves their KV cache. In each decoding step, a new token is generated based on the current token and the KV cache stored from earlier steps, continuing until a stop criterion is met.

## 4 Observations

In this section, we analyze the attention patterns during inference in long-context LLMs, providing insights that motivate our approach to transform the standard transformer into its corresponding hybrid variant. The study is conducted on the LLaMA3-8B-Instruct model (Dubey et al., 2024) using a sample from the LongBench (Bai et al., 2023) benchmark. Our key findings are as follows:

**Layer behavior in long-context LLMs during inference.** Previous research (Xiao et al., 2023) has shown that a large portion of attention in LLMs tends to focus on semantically unimportant tokens $X_{\text{initial}}$ (e.g., the first few tokens) and the most recent tokens $X_{\text{recent}}$ (i.e., tokens in the sliding window). We refer to this pattern as *lazy* behavior, likening it to skimming a paper by reading only the first lines and the conclusion. While it is also called attention sink (Xiao et al., 2023; Gu et al., 2024), we emphasize the shortcut nature by referring to it as lazy. Through our analysis, we find that even with long contexts, some layers exhibit more pronounced lazy behavior, which we define as *lazy layers*. The left panel of Figure 2 presents the attention patterns across different layers. We observe that some layers (e.g., layer 0) do not follow a clear pattern in

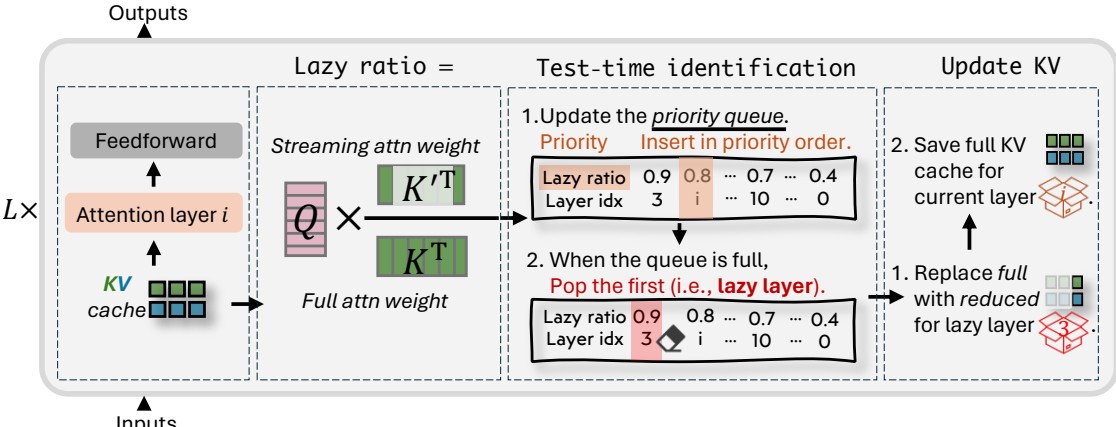

Figure 3: The framework of our LIGHTTRANSFER-TEST. A priority queue is maintained during the prefilling stage to store the lazy ratio and corresponding layer index after processing each layer. Once the queue reaches its capacity, the layer with the highest lazy ratio is identified as a lazy layer, and its KV cache is reduced, freeing memory for storing the KV cache of the current layer.

attention weight distribution, while others (e.g., layer 20) show a clear lazy behavior pattern. Consequently, a more memory-efficient attention mechanism can be employed in these lazy layers by retaining only a subset KV cache of constant size.

**Layer behavior remains consistent for a given input.** To further explore whether a layer consistently functions as a lazy layer during generation for a fixed prompt, we visualize the attention weights for $\{X_{\text{initial}}, X_{\text{recent}}\}$ across all layers for all generated tokens in the right panel of Figure 2, using a randomly selected sample (additional examples are provided in Figure 9). Notably, for a given input prompt, layers that exhibit lazy behavior maintain this pattern relatively consistently across tokens. This suggests a certain degree of stability in attention dynamics throughout the generation process. In addition, the indexes of these consistent lazy layers vary according to different prompts. This necessitates the test-time algorithm in the following section.

## 5 Methodology: LightTransfer

In this section, we introduce LIGHTTRANSFER, a method for converting pretrained transformers into hybrid architectures for a more efficient generation. LIGHTTRANSFER leverages our observation of lazy layers by replacing full attention with streaming attention. The method has two settings: (1) For tasks like long-context understanding, LIGHTTRANSFER-TEST allows for on-the-fly transformation at test time without requiring additional training. (2) For tasks demanding higher model capacity, such as o1-like long reasoning generation, LIGHTTRANSFER-TRAIN involves fine-tuning to adapt the model to the hybrid architecture.

### 5.1 LightTransfer-Test

As shown in Figure 3, the first step in applying LIGHTTRANSFER-TEST is identifying lazy layers, defined as those whose final $w_{\text{last}}$ number of tokens in queries (i.e., $X_{\text{last}}$) allocate the most attention to $X_{\text{initial}} \cup X_{\text{recent}}$. To measure how the model allocates attention at layer $i$, we define a lazy ratio $r_i$:

$$r_i = \frac{1}{w_{\text{last}}} \sum_{\hat{x} \in X_{\text{last}}} \sum_{x \in \{X_{\text{initial}}, X_{\text{recent}}\}} A_i(\hat{x}, x), \tag{1}$$

where $A_i(\hat{x}, x)$ is the averaged attention weight over all heads from a query token $\hat{x}$ to a key token $x$ at layer $i$. Intuitively, a higher $r_i$ indicates that $X_{\text{last}}$ focuses more heavily on these particular key sets, thus exhibiting more lazy attention. To ensure that only $P$ layers with the lowest lazy ratios maintain full attention during the prefilling stage and thus reduce peak memory usage, we adopt a priority queue. We treat the lazy ratio

Table 1: Torch style code for our lazy ratio calculation with flash attention.

```python
def Lazy_ratio_calculation(
    q, # bs * num_heads * seq_len * head_dim
    k, # bs * num_heads * seq_len * head_dim
    v, # bs * num_heads * seq_len * head_dim
    w_last, w_sink, w_recent):
    attn_out, lse = flash_attn(q, k, v,
causal=True, return_lse=True)
    q_last = q[:, -w_last:].permute(0, 2, 1, 3)
    k_comb = torch.cat([k[:, 0:w_sink],
k[:, -w_recent:]], dim=1).permute(0, 2, 3, 1)
    log_lazy_ratio = torch.matmul(q_last, k_comb)
.logsumexp(dim=-1)- lse
    return log_lazy_ratio
```

$r_i$ as the priority in a max-based priority queue of size $P$. Whenever the queue exceeds capacity, the layer with the highest lazy ratio is popped, labeled lazy, and its standard attention is replaced with streaming attention. Here we do not replace the standard attention with streaming attention in a head-wise manner due to the inefficiency, discussed in Appendix B.2. Specifically, for each lazy layer $i$, we retain only the KV caches corresponding to $\{X_{\text{initial}}, X_{\text{recent}}\}$ and discard others. During decoding, memory usage is naturally reduced because the decoding process relies on the already updated (and thus reduced) KV caches from the prefilling stage.

**Identification burden.** FlashAttention (Dao, 2023) is widely used to accelerate computations during the prefilling phase, but it does not explicitly expose attention weights. A direct application of our lazy layer identification strategy would thus require recomputing the attention matrix, incurring non-negligible overhead. To circumvent this issue, as shown in Table 1, we leverage the log-sum-exp values (i.e., the denominator) of all attention weights produced by FlashAttention. Consequently, we only need to recompute the streaming attention score (a constant-size matrix multiplication), thus eliminating the need for a full recomputation. Our identification algorithm mitigates additional latency introduced by full recomputation, resulting in only a slight throughput reduction of 0.0058 to 0.0014 relative to a baseline of 1 across sequence lengths from 4K to 32K. Notably, longer sequences result in smaller relative throughput reduction. This occurs because the prefill operation grows with sequence length, whereas our identification process remains $O(1)$. As a result, when $n$ is large, the identification overhead is overshadowed by the overall prefill cost.

## 5.2 LightTransfer-Train

For o1-like long reasoning tasks, where the input question typically consists of only a few dozen words, the lazy ratio $r_i$ is not a reliable indicator of *lazy*. Because the sliding window is relatively large compared to the input, $r_i$ remains at 1 across all layers. To address this, we adopt a pre-selection strategy. Specifically, for each sample in the training set, we feed both the question and the answer as input to the LLM, thereby providing sufficient context for each sample to reveal which layers are lazy. We then compute the frequency for each layer and select those with the highest lazy layer counts. However, frequency-based selection may not be fully optimal for each sample, while o1-like long reasoning tasks are inherently difficult, so additional fine-tuning allows the model to adapt to the new hybrid architecture and re-balance capacity across layers. Therefore, once these layers are identified, we perform supervised fine-tuning (SFT) under a hybrid architecture in which lazy layers employ streaming attention, while non-lazy layers retain standard attention. During inference, we simply rely on the preselected lazy layers, without requiring on-the-fly identification.

## 5.3 Theoretical Analysis

We first provide a theoretical analysis of the approximation error of LIGHTTRANSFER-TEST and then discuss how this analysis implies the performance of LIGHTTRANSFER-TRAIN. We would like to highlight that our lazy layer identification procedures in LIGHTTRANSFER-TEST are implicitly optimizing an upper bound of the error of the whole network output induced by reducing the KV cache. We denote the set of layer indexes whose KV cache is reduced as $\mathcal{I}$. For any layer $i \in \mathcal{I}$, we denote the attention score of the discarded KV pairs as $s_i = 1 - \sum_{x \in \{X_{\text{initial}}, X_{\text{recent}}\}} A_i(\hat{x}, x)$. Then we have the following upper bound of the error of the network output.

**Theorem 5.1** (Informal). *If the Frobenius norms of all the parameters in a $L$-layer with $H$-attention heads transformer are upper bounded by $B$ and the activation function is $L_{\mathsf{lip}}$-Lipschitz, then we have that*

$$\text{Err. of } \textsc{LightTransfer in logit}$$

$$\leq 2LB^2\big(H + L_{\mathsf{lip}}B + 4HB^2\big) + 2HB^2(1 + L_{\mathsf{lip}}B^2)\sum_{i \in \mathcal{I}} s_i.$$

*If we denote the error of hidden states at layer $i$ as $e_i$, then it evolves as*

$$e_i \leq e_{i-1} + C_1 \min\{2, C_2 \cdot e_{i-1}\} + 2H(B + L_{\mathsf{lip}}B^3)\mathbb{I}\{i \in \mathcal{I}\}s_i,$$

*where $C_1$ and $C_2$ are quantities related to $B$, $H$ and $L_{\mathsf{lip}}$.*

The formal statement and the proof of Theorem 5.1 are provided in Appendix F. We note that the error recursive expression consists of three terms. The first term represents the error from the previous layer. The second term represents the error from the previous layer amplified by the current layer. Thanks to the layer normalization, this term will be truncated by 2. The last term represents the newly introduced error if we shorten the KV cache at the current layer. By relaxing this recursive formula, we derive the upper bound of the error between the logits of our method and the original transformer. This shows that the error is upper bounded by the sum of the attention scores of the removed KV pairs up to an additive constant. We highlight that our algorithm optimizes Eqn. equation 1, which is exactly the upper bound of the error induced by LightTransfer in logit up to a constant. We note that this theorem also provides the error analysis of the initial point of this fine-tuning process. The fine-tuning will further decrease the error induced by LightTransfer shown in Theorem 5.1. This theoretical bound is further confirmed in Appendix B.3.

# 6 Experiments

In this section, we empirically validate that LightTransfer can accelerate LLM generation while maintaining long-text capabilities including two scenarios 1) long context understanding, and 2) o1-like long reasoning generation, and uncover several insightful findings.

## 6.1 Experiments on Long-Context Understanding Tasks

In these experiments, we only apply LightTransfer-Test. As previously discussed, the input length for these understanding tasks is sufficient to enable on-the-fly lazy-layer detection during the prefilling stage, making additional training unnecessary.

### 6.1.1 Experiments on LongBench

**Settings.** We evaluate LightTransfer-Test using four widely used LLMs, specifically LLaMA2-7B-chat (Touvron et al., 2023), Mistral-7B-Instruct (Jiang et al., 2023), LLaMA3-8B-Instruct and LLaMA3-70B-Instruct (Dubey et al., 2024) on LongBench (Bai et al., 2023), which is a multi-task benchmark designed to assess the long-context capabilities of LLMs. Detailed experimental configurations can be found in Appendix A. An ablation study on these hyperparameters is provided in the Appendix C.1.

**Baselines.** Since no existing approach can convert a transformer into a hybrid model at test time only, layer-level KV cache reduction methods serve as our closest baselines (Detailed discussions on how LightTransfer-Test relates to layer-level KV cache reduction methods are available in Appendix B.1). Specifically, we compare LightTransfer-Test against the following baselines: 1) Standard: a standard transformer-based model in which each layer employs the original self-attention mechanism. 2) Streaming LLM (Xiao et al., 2023): A memory-efficient approach that modifies each attention layer in a standard transformer to use only the KV cache for the first few tokens and the most recent tokens. 3) MiniCache (Liu et al., 2024a): An inter-layer KV cache reduction method that merges KV cache of every two adjacent layers after the model's midpoint using spherical interpolation while retaining important tokens to reduce cache storage. 4) SqueezeAttention (Wang et al., 2024c): An inter-layer KV cache reduction method that precisely distributes the KV-cache budget across layers.

**Results.** Table 2 summarizes the performance across various tasks in the LongBench (Bai et al., 2023) benchmark. We have the following findings:

Table 2: Performance comparison of LIGHTTRANSFER-TEST and baseline methods on LLaMA-2-7B-chat, Mistral-7B-Intruct, LLaMA-3-8B-Instruct, and LLaMA-3-70B-Instruct using LongBench. **Bold** denotes the best method, and underlined denotes the second best.

| | Single-Doc. QA | | | Muti.-Doc. QA | | | Summary | | | Few-shot | | | Syn. | | Code | | |
|---|---|---|---|---|---|---|---|---|---|---|---|---|---|---|---|---|---|
| | NrtvQA | Qasper | MF-en | HotpotQA | Musique | DuReader | GovReport | QMSum | MultiNews | TREC | TriviaQA | SAMSum | PCount | PRe | LCC | RB-P | Average |
| **_LLaMA2-7B-chat_** | | | | | | | | | | | | | | | | | |
| Standard | **19.1** | **21.6** | **36.9** | **27.7** | **8.6** | 6.5 | 27.1 | 20.8 | 26.0 | **64.0** | **83.6** | **41.3** | **2.9** | **7.5** | 60.6 | **54.9** | **31.8** |
| Streaming | 13.1 | 15.2 | 26.9 | 23.1 | 5.5 | 4.4 | 21.1 | 19.9 | 24.2 | 61.0 | 82.8 | 38.9 | 2.1 | 4.0 | 59.0 | 52.2 | 28.3 |
| MiniCache | 13.1 | 13.7 | 30.3 | 15.6 | 4.7 | **9.8** | 21.5 | **20.9** | 24.3 | 63.0 | 83.1 | 35.1 | 2.2 | 6.1 | 53.4 | 46.5 | 27.7 |
| SqueezeAtt. | 15.9 | 15.7 | 27.0 | 25.5 | 6.5 | 4.3 | 21.9 | 19.6 | 23.3 | 62.0 | 83.2 | 39.9 | 1.9 | 0.5 | 60.0 | 53.5 | 28.7 |
| LITTRANS | 15.8 | 18.3 | 30.1 | 27.3 | 7.0 | 4.7 | 22.7 | 20.2 | 25.1 | 62.0 | 82.8 | 39.6 | 2.1 | 1.2 | 59.4 | 53.6 | 29.5 |
| **_Mistral-7B-Instruct_** | | | | | | | | | | | | | | | | | |
| Standard | **29.7** | 40.5 | 53.4 | 50.0 | **29.1** | **32.9** | 34.9 | 25.4 | 27.7 | 76.0 | 89.1 | 47.3 | 5.0 | **98.5** | 60.4 | **62.1** | 47.6 |
| Streaming | 22.2 | 32.1 | 44.8 | 41.7 | 23.0 | 20.3 | 24.8 | 21.3 | 26.0 | 65.0 | 86.7 | 40.4 | 3.5 | 46.0 | 52.8 | 47.9 | 37.4 |
| MiniCache | 19.7 | 30.3 | 35.6 | 29.5 | 15.5 | 20.3 | 24.8 | 21.3 | 26.0 | 65.0 | 86.7 | 40.4 | 3.8 | 45.1 | 52.8 | 47.9 | 35.3 |
| SqueezeAtt. | 26.8 | 30.4 | 38.4 | 44.3 | 21.0 | 18.6 | 24.9 | 21.0 | 26.2 | 75.5 | 89.2 | 46.3 | **6.5** | 89.0 | 60.6 | 60.6 | 42.5 |
| LITTRANS | 29.0 | **41.0** | **53.6** | **50.5** | 27.5 | 32.3 | 34.8 | **25.4** | 27.3 | 76.0 | **89.3** | **47.3** | 6.0 | 97.5 | 59.9 | 61.3 | 47.4 |
| **_LLaMA-3-8B-Instruct_** | | | | | | | | | | | | | | | | | |
| Standard | **23.4** | **32.8** | **39.6** | **44.7** | 22.2 | 20.1 | 28.8 | 23.3 | 27.0 | 73.5 | 90.6 | 41.9 | 3.6 | **72.0** | 58.1 | 51.3 | **40.8** |
| Streaming | 19.5 | 17.5 | 26.1 | 36.4 | 16.1 | 12.1 | 22.8 | 21.4 | 25.4 | 66.0 | 86.4 | 40.1 | 3.5 | 70.7 | 59.7 | 54.2 | 36.1 |
| MiniCache | 17.4 | 10.9 | 18.4 | 11.5 | 6.7 | 15.9 | 23.8 | 20.1 | 25.5 | 74.5 | 84.5 | 37.4 | 3.2 | 64.1 | 48.5 | 45.3 | 31.7 |
| SqueezeAtt. | 20.0 | 19.6 | 26.2 | 37.5 | 18.7 | 13.3 | 23.8 | 22.0 | 23.8 | 72.5 | 90.0 | 41.5 | 6.7 | 66.0 | 55.2 | 47.6 | 36.5 |
| LITTRANS | 23.2 | 18.3 | 35.7 | 43.7 | 20.9 | 14.5 | 24.1 | 22.3 | 26.0 | 71.0 | **91.1** | 41.4 | **6.9** | 67.0 | **60.2** | 53.4 | 38.7 |
| **_LLaMA-3-70B-Instruct_** | | | | | | | | | | | | | | | | | |
| Standard | 25.6 | **46.4** | 51.4 | 49.8 | 28.8 | **28.7** | 32.2 | 22.4 | 27.6 | 73.5 | 92.9 | 45.7 | **12.0** | 68.5 | 41.6 | 69.7 | **44.8** |
| Streaming | 25.4 | 36.2 | 34.4 | 44.3 | 22.7 | 15.0 | 25.8 | 20.2 | 26.2 | 66.5 | 91.1 | 43.6 | 11.5 | 68.0 | 41.9 | 67.1 | 40.0 |
| MiniCache | 25.1 | 45.2 | 38.4 | 46.2 | 24.9 | 17.8 | 29.1 | 22.3 | 27.1 | 71.0 | 86.7 | 41.3 | 10.1 | 67.0 | 35.6 | 54.4 | 40.1 |
| SqueezeAtt. | **26.3** | 36.8 | 34.0 | 48.1 | 25.0 | 17.5 | 28.0 | 21.5 | 25.5 | 71.5 | 92.8 | 44.8 | 11.5 | 67.0 | 41.5 | 68.5 | 41.3 |
| LITTRANS | 25.8 | 44.3 | 46.9 | 49.3 | **29.4** | 20.8 | 28.4 | 22.1 | 26.9 | **74.0** | 92.3 | 43.9 | 11.5 | 68.0 | **43.6** | **69.8** | 43.6 |

_LLMs exhibit redundancy across layers._ As shown in the table, although MiniCache has some limitations, both SqueezeAttention and LIGHTTRANSFER-TEST enable the model to handle long-text tasks effectively, incurring only a slight performance decrease (an average drop of 4.0% and 1.5%, respectively) when removing the KV cache in 50% of the layers. This finding suggests that LLMs exhibit redundancy in their layer-level KV caches.

_The transferred hybrid architectures can preserve strong long-context understanding capability._ LIGHTTRANSFER-TEST applies streaming attention in some layers of a transformer-based model while retaining standard self-attention in others, striking an effective balance between computational efficiency and representational capacity. In contrast, MiniCache adopts cross layer attention (CLA) (Brandon et al., 2024) (sharing one KV cache across adjacent layers), and SqueezeAttention allocates distinct KV-cache quotas per layer. Under a higher compression ratio than MiniCache and the same ratio as SqueezeAttention, LIGHTTRANSFER-TEST surpasses them by 6.1% and 2.6%, respectively, demonstrating the effectiveness of transitioning transformers into hybrid models for memory-efficient inference. This superiority partially originates from the fact that our algorithm explicitly optimizing the error upper bound in Theorem 5.1. In contrast, the optimization methods of MiniCache and SqueezeAttention do not control the error induced by KV reduction in a theoretically plausible manner.

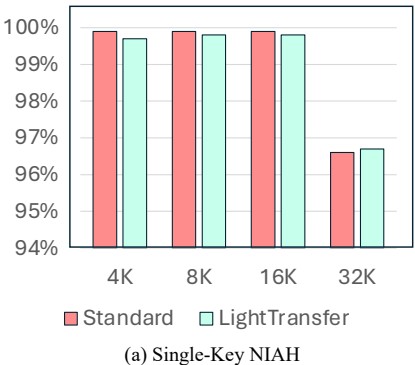 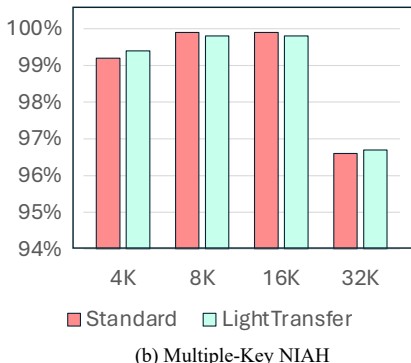

(a) Single-Key NIAH  (b) Multiple-Key NIAH

Figure 4: Performance comparison of LIGHTTRANSFER and standard model on NIAH tasks using Mistral-7B-Instruct.

Table 3: Performance comparison of LIGHTTRANSFER-TRAIN and baseline methods on three mathematical benchmarks using QwQ-32B. **Bold** denotes the best method, and underlined denotes the second best.

| Method | MATH-OAI | AIME24 | GSM8K |
|---|---|---|---|
| QwQ-STILL | 90.2 | 46.7 | **95.6** |
| LongGen | 78.2 | 16.7 | 95.4 |
| LITTRANS | **90.7** | **53.3** | 95.5 |

### 6.1.2 Experiments on NIAH

**Settings.** We also evaluate whether LIGHTTRANSFER-TEST can preserve in-context retrieval capabilities while replacing some standard attention layers into memory-efficient streaming attention. The evaluation is conducted on single-key and multiple-key NIAH tasks collected in the Ruler (Hsieh et al., 2024) benchmark. We report the performance with input context lengths of 4K, 8K, 16K, and 32K. Detailed experimental configurations can be found in Appendix A.

**Results.** Figure 4 summarizes the performance on NIAH tasks, with the context length ranging from 4K to 32K. While our LIGHTTRANSFER-TEST replacing select transformer layers with streaming attention reduces memory overhead, strategically retaining original attention mechanisms in deeper layers ensures robust long-range dependency modeling. This explains the maintained performance on single-key tasks (32K: 96.7% vs standard 96.6%) and competitive multi-key results at 32K (78.2% vs 78.9%). The retained standard layers serve as an anchor for cross-token reasoning, which is crucial for in-context retrieval.

### 6.2 Experiments on o1-like Long Reasoning Tasks

In these experiments, we investigate the effectiveness of LIGHTTRANSFER-TRAIN on o1-like long reasoning generation tasks. While these tasks feature relatively short inputs, they demand intricate reasoning. Consequently, we SFT the model with approximately 5K training examples to facilitate swift adaptation within the transferred hybrid architecture.

**Settings.** Experiments are conducted on three widely used mathematical benchmarks AIME24, MATH-OAI, and GSM8K. We use greedy decoding to evaluate the performance of our model with maximum tokens set to 32K. We aim to experiment with models capable of generating CoT, such as QwQ. However, as the original training data for QwQ is not publicly available, we adopt the public training set provided by QwQ-STILL (Min et al., 2024), a model demonstrated to achieve comparable performance to QwQ. Specifically, following QwQ-STILL, we apply a simple distillation approach using Qwen2.5-32B-Instruct as our base model. We generally follow the original training set of QwQ-STILL, and replace 50% of layers with streaming attention. To mitigate the training complexity of attention, we optimize LIGHTTRANSFER-TRAIN training using Flex Attention (Dong et al., 2024).

Table 4: Performance comparison of LIGHTTRANSFER and baseline methods that operate without tuning LLM parameters on mathematical benchmarks using QwQ-32B. **Bold** denotes the best method.

| Method | MathOAI | AIME24 |
|---|---|---|
| QwQ-STILL | 90.2 | 46.7 |
| MiniCache-Test | 20.0 | 0.0 |
| SqueezeAttention-Test | 82.6 | 15.0 |
| DuoAttn.-Train | 79.2 | 33.3 |
| LIGHTTRANSFER-TEST (OURS) | 85.0 | 40.0 |
| LIGHTTRANSFER-TRAIN (OURS) | **90.7** | **53.3** |

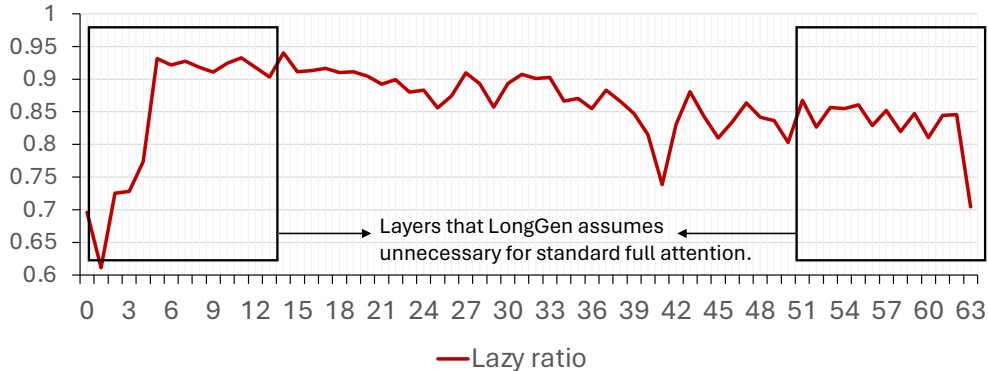

Figure 5: Lazy ratio scores across layers in QwQ-32B-STILL.

**Baselines.** We compare our LIGHTTRANSFER-TRAIN against the following baselines: 1) QwQ-STILL (Min et al., 2024): a distilled model on Qwen2.5-32B-Instruct that achieves performance comparable to QwQ-32B-Preview, whose training data is publicly available. 2) LongGen (Ge et al., 2024): an approach that assumes the layers at both ends of the model do not handle global information and predefines the replacement of those layers with sparse attention.

We further benchmark LIGHTTRANSFER-TEST against (i) existing *training-free* sparsification baselines, and (ii) DuoAttention (Xiao et al., 2024). Unlike our approach, DuoAttention must first run on a *separate calibration set* to pinpoint those attention heads whose KV cache should be evicted, whereas LIGHTTRANSFER operates entirely *calibration-free.*

**Results.** Table 3 shows that LIGHTTRANSFER-TRAIN retains its performance on Math-OAI (+0.5%), AIME24 (+6.6%) and GSM8K (-0.1%). In contrast, LongGen, which assumes its middle layers require standard attention, exhibits no drop on GSM8K but suffers a 30.0% and 12.4% decrease on AIME24 and Math-OAI, respectively. While the unchanged GSM8K results for LongGen may indicate that GSM8K poses lower complexity for these models, the broader comparisons nevertheless highlight the strength of our data-driven layer selection. Specifically, our LIGHTTRANSFER-TRAIN calculates each layer's *lazy ratio* (Figure 5) and replaces those exhibiting the highest, which is proven to be more robust than hand-crafted assumptions. Moreover, our findings underscore the existence of layer-level KV cache redundancy even in o1-like long reasoning models, emphasizing the promise of hybrid transformer architectures. Meanwhile, as shown in Table 4, LIGHTTRANSFER-TEST surpasses all other training-free baselines on both MathOAI and AIME24, highlighting its suitability for long reasoning generation tasks.

### 6.3 Ablation Studies & Analysis

**Standard layer retention ratio vs. model performance.** As shown in Figure 6, we systematically vary the fraction of layers that use original attention from 0.25 to 0.5, up to 0.75, for both LLaMA3-8B-Instruct and LLaMA3-70B-Instruct on LongBench benchmark. As expected, higher retention ratios consistently

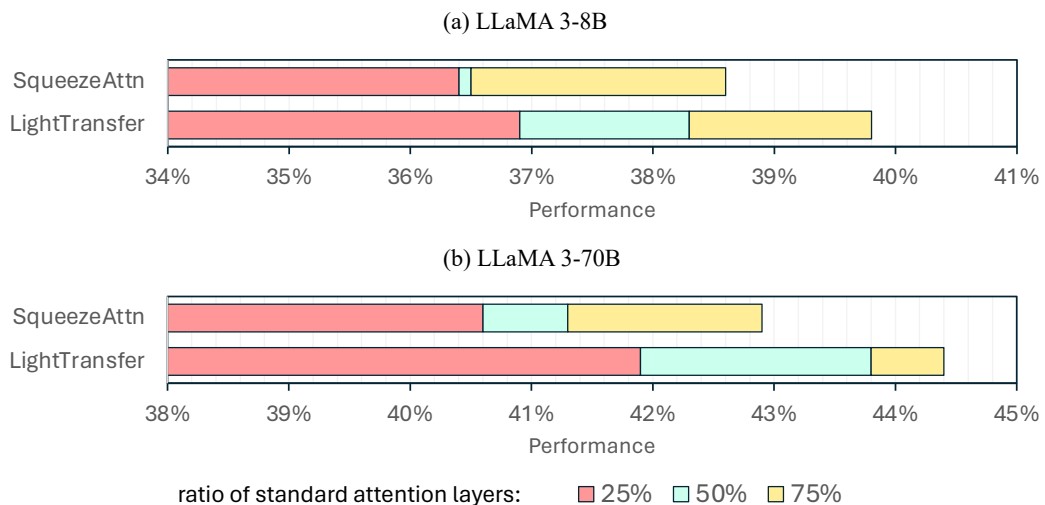

Figure 6: Effect of retaining standard attention in more layers on LongBench.

Table 5: Relative token-generation throughput at different sequence lengths (4K, 8K, 16K, and 32K) compared to the Full baseline. **Bold** denotes the best method.

| Method | 4K | 8K | 16K | 32K |
|---|---|---|---|---|
| SqueezeAtten. | 1.03× | 1.09× | 1.12× | 1.04× |
| MiniCache | 1.26× | 1.29× | 1.52× | 1.41× |
| LIGHTTRANSFER | **1.44×** | **1.78×** | **2.17×** | **1.75×** |

yield improved model performance. However, this comes at the cost of increased memory consumption, highlighting the trade-off between efficiency and accuracy. Notably, across all compression settings examined, LIGHTTRANSFER-TEST surpasses the strongest baseline on that benchmark (i.e., SqueezeAttention), thereby underscoring the benefit of transitioning standard transformers to hybrid models via strategical designs for more efficient generation.

**Throughput of token generation.** To evaluate how these memory optimizations impact token-generation throughput, we conduct experiments with Mistral-7B on the Ruler benchmark under maximum batch-size configurations. Input sequence lengths of 4K, 8K, 16K, and 32K were tested while retaining 50% of the standard attention layers. As shown in Table 5, LIGHTTRANSFER-TEST consistently achieves the highest throughput compared with other training-free test time inter-layer KV cache reduction methods. In contrast, SqueezeAttention, despite having the same compression ratio, fails to reduce peak memory usage during the prefilling phase, since it must complete prefilling for all layers before applying compression. This constraint limits the feasible batch size, restricting potential throughput. Meanwhile, MiniCache exhibits lower throughput due to its smaller compression ratio (i.e., removing KV caches in at most 25% of layers). These findings underscore the effectiveness of LIGHTTRANSFER in balancing memory usage and computational efficiency.

**Effect of different layer replacement strategies.** As shown in Figure 7 (a-d), we experiment with four different layer replacement strategies for integrating memory-efficient streaming attention into transformers, with consistent replacement counts (except Standard). The results shown in Figure 7 indicate noticeable reductions for Pyramid and Random strategies, suggesting that the predefined expectations about each layer's function may not fully align with their actual roles. Moreover, the performance of our LIGHTTRANSFER surpasses other strategies, suggesting that LIGHTTRANSFER is effective in reducing memory usage while maintaining performance. Additional results with two extra baselines are provided in Appendix C.5.

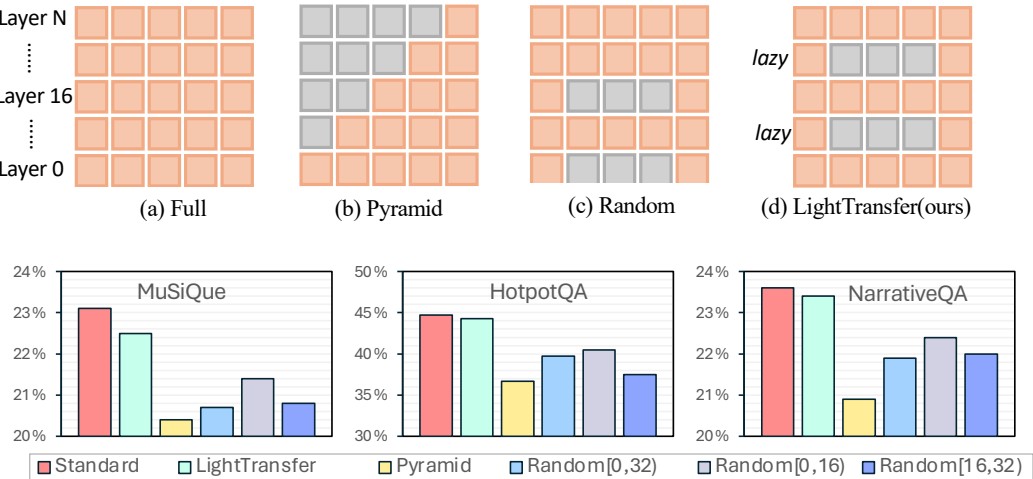

Figure 7: Different layer replacement strategies and their performance on LLaMA3-8B-Instruct: 1) Standard: Use standard attention in all layers. 2) Our LIGHTTRANSFER: Dynamically identify lazy layers on the fly, and replace their attention mechanism accordingly. 3) Pyramid: Replace each layer with memory-efficient attention; the budget decreases with depth, forming a pyramid-like structure. 4) Random: Randomly replace layers with memory-efficient attention within the ranges $[0, 16)$, $[16, 32)$, or $[0, 32)$. We keep a **same** number of replaced layers, except Standard.

## 7 Conclusion

We present LIGHTTRANSFER, a lightweight framework for transforming standard transformers into hybrid models for more efficient generation by identifying *lazy* layers and replacing their full-attention modules with streaming attention. Extensive experiments show that even when half of the transformer layers are replaced with streaming attention, LIGHTTRANSFER delivers up to a $2.17\times$ increase in throughput while incurring less than a $1.5\%$ performance drop on LongBench. For advanced long reasoning generation tasks like AIME24, our method achieves these gains without any performance degradation on QwQ-STILL.

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

# A Settings.

We adopt a generative format where answers are produced using greedy decoding for all tasks. All the experiments are conducted using NVIDIA A100 using PyTorch and HuggingFace Transformers replaced with `flash_attention_with_kvcache` to accelerate. All model weights, activations, and KV caches use BF16 precision, with no quantization applied. We set the sink token num $w_{\text{sink}} = 4$ and the window size $w_{\text{recent}} = 1020$.

## A.1 Settings on LongBench

The input context window sizes of LLaMA2-7B-chat, Mistral-7B-Instruct, LLaMA3-8B-Instruct and LLaMA3-70B-Instruct are 4K, 8K, and 32K, with average tokenized sequence lengths approximately 13K, 12K, 10K, and 10K in LongBench. For evaluation, we use the metrics recommended by LongBench. Due to space constraints, we only include the performance of 16 randomly selected tasks out of the 21 LongBench tasks. For MiniCache, as the code was not open-sourced before our submission, we reimplemented it based on the original paper and the SLERP (Shoemake, 1985) code it references. We followed all the hyper-parameters outlined in the paper, except for the number of retention tokens. SqueezeAttention and our LIGHTTRANSFER-TEST TIME are both set to the same compression ratio, equivalent to removing KV caches from 50% of the layers (i.e., $P$ is set to 50% of the total number of layers), whereas MiniCache is set to 25% (i.e., its maximum possible compression).

## A.2 Settings on NIAH

The evaluation is conducted using the metrics recommended by Ruler. Because synthetic in-context retrieval tasks in the Ruler benchmark require more extensive global context, we use a slightly lower removal ratio here than the one applied to LongBench. In our LIGHTTRANSFER-TEST TIME setup, we remove the KV caches from 25% of the layers.

# B Discussions

## B.1 Relationships with Test-time KV Cache Reduction.

Some techniques (Xiao et al., 2023; Li et al., 2024b; Wang et al., 2024b; Zhang et al., 2024e; Liu et al., 2024b; Yang et al., 2024; Zhang et al., 2024d) identify redundant tokens within each attention layer and

evict their associated KV cache at test time, thereby effectively lowering memory usage. Within this line of research, the approach most closely aligned with our LIGHTTRANSFER-TEST TIME specifically targets layer-level KV cache redundancies during inference, aiming to further optimize memory consumption by examining how different layers store and reuse keys and values. However, current methods only consider the relationships of KV caches across layers from a relatively coarse perspective for reducing KV caches across layers. For example, MiniCache Liu et al. (2024a) focuses on the similarity of KV caches between layers, while SqueezeAttention Wang et al. (2024c) optimizes cache usage without a detailed investigation into the internal mechanisms of transformers. In contrast, our LIGHTTRANSFER approach goes further by examining how each layer functions and selectively replacing certain layers with more memory-efficient architectures.

## B.2 Why We Do Not Adopt a Head-Wise Hybrid Model

Prior studies typically do not consider a head-wise hybrid design Lieber et al. (2024); Gemma et al. (2024); Sun et al. (2024); Botev et al. (2024); De et al. (2024). One practical reason is that LLMs often employ tensor parallelism (TP) to distribute computation across multiple GPUs. In this setup, a single layer generally contains multiple attention heads (e.g., eight heads per layer), and each head is handled by a separate GPU. If different heads in the same layer maintain different KV cache sizes, GPUs with smaller caches must wait for those with larger caches to finish. This synchronization bottleneck cancels out any latency benefits gained from compressing only certain heads, making a head-wise hybrid approach inefficient in real-world deployments.

In addition, when using TP only, we observe that the head-wise hybrid model struggles to improve throughput. To make a fair comparison, we also implement both data-parallel attention + TP feed-forward network (DP+TP) method based on the state-of-the-art inference framework SGLang, and evaluate this strategy with pure TP on LLaMA3-70B using an 8×A100 40G node.

Despite idealized settings (identical input lengths and perfect batch distribution with $BSZ\%GPU_{num} = 0$), DP+TP significantly reduces throughput—only **0.0735×** that of TP—and reduces the maximum supported sequence length to only 1/128×. The root cause is that TP shards attention layer parameters across GPUs, while DP+TP replicates them, consuming an additional 157.5 GB of GPU memory. Moreover, TP shards the KV cache *per request*, allowing it to support 8× longer context lengths.

Table 6: Throughput comparison (tokens/s) for LLaMA3-70B under different sequence lengths. DP+TP suffers from severe memory issues.

| Length | 512k | 32k | 16k | 8k | 4k |
|---|---|---|---|---|---|
| DP+TP (tokens/s) | OOM | OOM | OOM | OOM | 145.6 |
| TP (tokens/s) | 22.6 | 360.0 | 772.1 | 1188.6 | 1981.9 |

Another critical issue with DP+TP is its difficulty in achieving load balance, a concern also raised in the DeepSeekV3R1 Inference Report. In extreme imbalance cases, throughput under DP+TP decreases linearly with GPU count. While increasing batch size may alleviate this in short-context scenarios, it does not hold for long-context settings, where KV cache memory constraints restrict batch size (e.g., only 16 sequences at 32k tokens, even with TP).

Furthermore, in production deployments with *prefill-decode separation (PD)*, DP+TP schedules workload balancing based on prefill lengths. We argue this heuristic fails in many real-world settings:

- **Multi-turn conversations**: Response lengths in later turns are highly unpredictable, violating assumptions of balanced prefill length.

- **Long-form generation**: Chain-of-Thought models may generate over 16K tokens even from short prompts, again breaking static prefill balancing.

Finally, head-wise hybrid methods are hard to integrate with existing inference systems like vLLM and SGLang due to KV cache granularity constraints. In contrast, industry-leading pretrained hybrid models (e.g., Gemma-2/3) adopt a **layer-wise hybrid structure**, validating our design decision.

Table 7: Performance under different hyperparameters.

(a) Window size $w_{recent}$

| Window size | 252 | 508 | 1020 | 2044 |
|---|---|---|---|---|
| **Performance** | 39.5 | 39.8 | 39.8 | 40.1 |

(b) Sink token count $w_{sink}$

| Sink num | 0 | 2 | 4 | 6 |
|---|---|---|---|---|
| **Performance** | 26.5 | 39.8 | 39.8 | 39.9 |

(c) $w_{last}$

| $w_{last}$ | 8 | 16 | 32 | 64 |
|---|---|---|---|---|
| **Performance** | 39.9 | 39.8 | 39.9 | 39.7 |

### B.3 Additional Evidence for the Usefulness of the Bound

We measure the divergence between the output distributions of the standard and LIGHTTRANSFER methods after prefilling with identical real-world samples. We employ the *Kullback–Leibler (KL) divergence* as a principled metric to quantify distributional discrepancy.

Table 8: KL divergence on seven LongBench datasets using LLaMA3-8B-Instruct. Lower values indicate closer alignment.

| Dataset | No layer reduction | 50% layer reduction |
|---|---|---|
| 2WikiMultihopQA | $-4 \times 10^{-6}$ | $-9 \times 10^{-6}$ |
| HotpotQA | $-6 \times 10^{-6}$ | $-1 \times 10^{-5}$ |
| MultiFieldQA-EN | $-4 \times 10^{-6}$ | $-8 \times 10^{-6}$ |
| MultiFieldQA-ZH | $-1.1 \times 10^{-5}$ | $-1.6 \times 10^{-5}$ |
| MuSiQue | $-6 \times 10^{-6}$ | $-1 \times 10^{-5}$ |
| NarrativeQA | $-1.1 \times 10^{-5}$ | $-1.5 \times 10^{-5}$ |
| Qasper | $-6 \times 10^{-6}$ | $-1 \times 10^{-5}$ |
| **Average** | $\mathbf{-7.1 \times 10^{-6}}$ | $\mathbf{-1.2 \times 10^{-5}}$ |

The extremely small KL values, consistently approaching zero across diverse datasets, confirm the practical tightness of our theoretical bound. Furthermore, the KL divergence values increase with the reduction of layers, consistent with our theoretical predictions. Such negligible divergence indicates that LIGHTTRANSFER faithfully preserves the original token-level predictive distribution during prefilling.

### B.4 Limitations

Firstly, we acknowledge that the performance of the LightTransfer method in the test setting does not match its training performance. Improving the effectiveness of this LightTransfer-Test remains an important direction for future research. Secondly, we recognize that our training method constitutes fine-tuning, and thus does not fully exploit the potential of the hybrid model architecture. Due to constraints in available GPU resources and data, we were unable to comprehensively explore a fully native hybrid model. We believe that future work with enhanced computational resources could unlock additional capabilities inherent in this promising hybrid model structure.

# C  Additional Experiment Results.

## C.1  Impact of Hyperparameters

We adopt these hyperparameters either directly from StreamingLLM Xiao et al. (2023) (i.e., $w_{\text{sink}}$ and $w_{\text{recent}}$), ensuring consistency with established practices in the field, or through preliminary experiments (i.e., $w_{\text{last}}$). We also conducted additional experiments to analyze the impact of hyperparameters ($w_{\text{sink}}$, $w_{\text{recent}}$, and $w_{\text{last}}$) on model performance. As shown in Table 7, the variation in performance remains within one percentage point across different configurations, demonstrating the robustness of our approach to hyperparameter choices.

## C.2  Combination with Intra-layer KV Cache Reduction Methods

To illustrate the orthogonality between our LIGHTTRANSFER-TEST and intra-layer KV cache compression methods, we conduct additional experiments that combine LIGHTTRANSFER-TEST with SnapKV (a cutting-edge method for intra-layer KV cache reduction). In these experiments, SnapKV is applied to compress the KV cache in non-lazy layers, while LIGHTTRANSFER-TEST remains active for lazy layers. We use Qwen2.5-3B-chat-32K for this analysis. As shown in Figure 8, leveraging LIGHTTRANSFER-TEST alongside an intra-layer KV cache compression method can further reduce KV cache size while preserving model performance, underscoring LIGHTTRANSFER-TEST's orthogonality to existing methods focused on intra-layer redundancies.

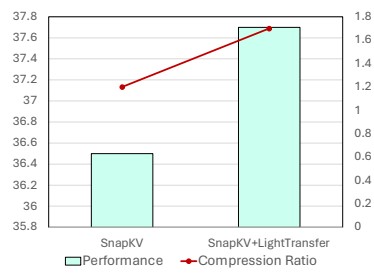

Figure 8: Comparison of SnapKV and SnapKV+LightTransfer.

## C.3  Comparison with Head-wise KV Cache Reduction Methods

We use 50% sparsity across all methods and evaluate on LLaMA3-8B-Instruct-Gradient-1048K to align with DuoAttention's released checkpoint.

Under the same sparsity level, our method is on par with DuoAttention (Xiao et al., 2024), which requires training and is head-wise. FastGen (Ge et al., 2023) will be OOM on most datasets because it is incompatible with flash attn (Xiao et al., 2024). RazorAttn (Tang et al., 2024) and HeadKV (Fu et al., 2024) are highly sensitive to the calibration set. Although we follow their reported calibration setup, the results are hard to reproduce. Therefore, we use the full LongBench dataset (input only) for their calibration. Unlike (Xiao et al., 2024; Ge et al., 2023; Tang et al., 2024; Fu et al., 2024), our search strategy only needs one additional matmul operation, which is fast. Thus, LightTransfer does not need any calibration set and can perform on-the-fly search, thereby avoiding sensitivity issues.

## C.4  Performance on MoE architectures

We conducted additional experiments using the Qwen1.5-MoE-14.3B-A2.7B (Qwen, 2024a) model on tasks within LongBench. As shown in Table 10, when replacing 50% of its layers with streaming attention, LightTransfer exhibits a performance drop of less than 1% on the MoE architecture, outperforming other layer-wise KV cache pruning baselines. This observation is consistent with our findings on non-MoE transformer models, thereby further confirming the robustness and effectiveness of our approach.

## C.5  Comparasion with other layer layer replacement strategies

To further study the effectiveness of our approach, we benchmark it against the following two baselines :

(i) **Shapley value–based** (Zhang et al., 2024c), which replaces layers according to their estimated Shapley contribution;

(ii) **BERTology** (Rogers et al., 2021), which iteratively substitutes the least influential layers.

Table 9: Comparison with Head-wise KV Cache Reduction Methods on LLaMA3-8B-Instruct-Gradient-1048K.

| Dataset | Razor | HeadKV | Ours | DuoAttn |
|---|---|---|---|---|
| Training Free | Yes | Yes | Yes | No |
| qasper | 19.69 | 29.97 | 26.85 | 27.02 |
| multiqa_en | 27.62 | 30.93 | 48.26 | 53.69 |
| hotpotqa | 23.98 | 26.96 | 37.07 | 35.52 |
| 2wiki | 24.83 | 25.70 | 30.41 | 28.08 |
| multinews | 25.84 | 27.31 | 26.50 | 27.76 |
| trec | 55.50 | 58.50 | 66.00 | 69.00 |
| triviaqa | 63.74 | 78.54 | 87.68 | 87.32 |
| samsum | 40.10 | 39.70 | 41.04 | 41.13 |
| pcount | 2.26 | 0.39 | 2.00 | 2.00 |
| lcc | 32.19 | 34.54 | 41.20 | 39.24 |
| repo-p | 32.15 | 36.06 | 40.02 | 40.04 |
| **Average** | **31.63** | **35.33** | **40.64** | **40.98** |

Table 10: Performance on Qwen1.5-MoE-14.3B-A2.7B.

| Dataset | Standard | MiniCache | SqueezeAttn | LitTrans |
|---|---|---|---|---|
| qasper | 30.19 | 18.66 | 23.48 | 26.92 |
| multifieldqa_en | 38.47 | 24.14 | 29.58 | 37.23 |
| hotpotqa | 10.17 | 5.57 | 8.56 | 9.12 |
| 2wikimqa | 11.92 | 7.64 | 11.35 | 12.78 |
| multi_news | 24.77 | 20.09 | 20.94 | 24.63 |
| trec | 68.00 | 66.00 | 64.50 | 65.50 |
| triviaqa | 86.35 | 70.44 | 84.80 | 85.35 |
| samsum | 38.09 | 24.87 | 38.22 | 37.74 |
| passage_count | 1.67 | 2.04 | 3.30 | 3.10 |
| lcc | 48.33 | 33.38 | 44.90 | 48.68 |
| repobench-p | 40.89 | 22.09 | 36.62 | 38.33 |
| **Average** | **36.26** | **26.81** | **33.30** | **35.43** |

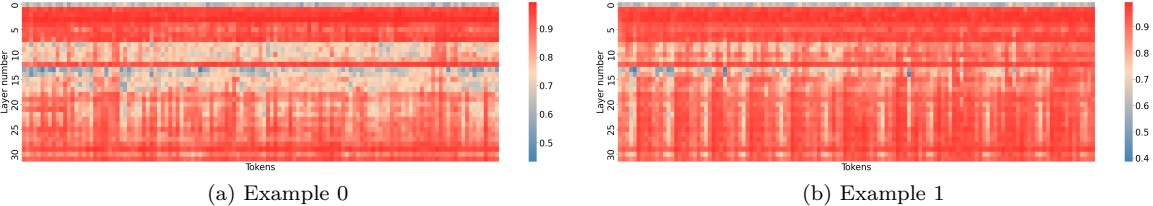

| (a) Example 0 | (b) Example 1 |

Figure 9: Additional examples of layer behavior across tokens.

Table 11 summarises the performance on three tasks from LongBench. Our **LightTransfer** strategy attains the best performance on all datasets, surpassing the next-best baseline by up to +3.7% (HotpotQA) while maintaining the same inference budget.

Table 11: Comparison of different layer-replacement strategies on three datasets on LLaMa3-8B-Instruct.

| Method | HotpotQA | MuSiQue | NarrativeQA |
|---|---|---|---|
| Shapley value–based | 39.97 | 18.18 | 19.63 |
| BERTology | 42.48 | 18.13 | 20.91 |
| LightTransfer | **43.70** | **20.90** | **23.20** |

## D  More Examples

### D.1  Examples about Layer Behavior across Tokens

Additional examples of layer behavior across tokens for a given input can be found in Figure 9. The examples are randomly chosen from LongBench benchmarks. The analysis is conducted using LLaMA3-8B-Instruct.

## E  Notation

For a positive integer $N \in \mathbb{N}$, we define the set $[N] = \{1, \cdots, N\}$. For a vector $x \in \mathbb{R}^d$, we adopt $\| \cdot \|_p$ to denote the $\ell_p$ norm of vectors. For a matrix $X = [x_1^\top, \cdots, x_{d_1}^\top]^\top \in \mathbb{R}^{d_1 \times d_2}$, where $x_i \in \mathbb{R}^{d_2}$ for $i = 1, \cdots, d_1$, we define the $\ell_{p,q}$-norm of $X$ as $\|X\|_{p,q} = \| [\|x_1\|_p, \cdots, \|x_{d_1}\|_p] \|_q$, i.e., we first apply $\ell_p$ norm in a row-wise manner and then apply $\ell_q$ norm. The Frobenius norm $\| \cdot \|_{2,2}$ is also denoted as $\| \cdot \|_{\mathbf{F}}$. For a matrix $X \in \mathbb{R}^{a \times b}$, its $i$-th row and $i$-th column are denoted as $[X]_{i,:}$ and $[X]_{:,i}$, respectively. The element at $i$-th row and $j$-th column of $X$ is denoted as $[X]_{i,j}$.

## F  Theoretical Analysis

In this section, we provide the theoretical analysis of the proposed method. We first define the transformer structure we analyze in this paper. In fact, we analyze the LLaMA-type structure (Dubey et al., 2024), i.e., the transformers that adopt the pre-norm and the res-link. The input of the transformer is the embedding of the tokens $X \in \mathbb{R}^{N \times d}$, where $N$ is the number of tokens, and $d$ is the dimension of the token embedding. We consider a $L$-layer transformer, i.e., there are $L$ transformer blocks in the network. Each transformer block consists of a Multi-Head Attention (MHA) and a Feed-Forward (FF) module. The MHA module is a combination of multiple causal self-attention modules. Each causal self-attention module is defined as

$$\mathsf{attn}(X, W_Q, W_K, W_V) = \mathsf{softmax}(X W_Q W_K^\top X^\top + M) X W_V,$$

where $X \in \mathbb{R}^{N \times d}$ is the input, $W_Q, W_K \in \mathbb{R}^{d \times d_k}$ and $W_V \in \mathbb{R}^{d \times d}$ are the weights of the self-attention module, and $M \in \mathbb{R}^{N \times N}$ is the causal mask. The causal mask is defined as

$$[M]_{i,j} = \begin{cases} 0 & \text{if } i \geq j \\ -\infty & \text{otherwise.} \end{cases}$$

The MHA with $H$ heads is defined as

$$\mathsf{mha}\Big(X, \{W_{Q,h}, W_{K,h}, W_{V,h}\}_{h=1}^{H}\Big) = \sum_{h=1}^{H} \mathsf{softmax}(X W_{Q,h} W_{K,h}^{\top} X^{\top} + M) X W_{V,h},$$

where $X \in \mathbb{R}^{N \times d}$ is the input, $W_{Q,h}, W_{K,h} \in \mathbb{R}^{d \times d_k}$ and $W_{V,h} \in \mathbb{R}^{d \times d}$ are the weights of the $h$-th head of MHA. Here we just merge the parameter $W_O$ into $W_V$ for ease of notation. Our analysis can be directly applied to the parameterization that explicitly includes $W_O$ as a weight. The FF module applies transformations to $X$ in a row-wise manner, which can be defined as

$$\mathsf{ffn}(X, W_{A,1}, W_{A,2}) = \sigma(X W_{A,1}) W_{A,2},$$

where $W_{A,1}, W_{A,2} \in \mathbb{R}^{d \times d}$ are weights of FF module, and $\sigma(\cdot)$ is an element-wise activation function. For example, $\sigma$ can be $\mathsf{ReLU}$ function. We require that $\sigma$ is a Lipschitze function.

**Assumption F.1.** The activation function $\sigma(\cdot)$ is $L_{\mathsf{lip}}$-Lipschitze, i.e., $|\sigma(x) - \sigma(y)| \leq L_{\mathsf{lip}}|x - y|$ for any $x, y \in \mathbb{R}$.

We note that this assumption is satisfied by all the popular activation functions, including ReLU, sigmoid, ELU, and GELU. The input of the transformer is denoted as the output of the 0-th layer, i.e., $X^{(0)} = X$. Then the $i$-th block processes in the input $X^{(i-1)}$ as

$$Y^{(i)} = X^{(i-1)} + \mathsf{mha}\Big(\mathsf{LN}(X^{(i-1)}), \{W_{Q,h}^{(i)}, W_{K,h}^{(i)}, W_{V,h}^{(i)}\}_{h=1}^{H}\Big) \tag{2}$$

$$X^{(i)} = Y^{(i)} + \mathsf{ffn}\big(\mathsf{LN}(Y^{(i)}), W_{A,1}^{(i)}, W_{A,2}^{(i)}\big), \tag{3}$$

where the superscript $(i)$ denotes the parameters and hidden states at layer $i$, and $\mathsf{LN}$ is the row-wise normalization of the input. To simplify the mathematical calculation, we defined $\mathsf{LN}$ as

$$\mathsf{LN}(x) = \begin{cases} x & \text{if } \|x\|_2 \leq 1 \\ x/\|x\|_2 & \text{otherwise} \end{cases}$$

Our analysis can be directly applied to the LayerNorm function of PyTorch. For ease of notation, we will abbreviate $\mathsf{mha}(\cdot, \{W_{Q,h}^{(i)}, W_{K,h}^{(i)}, W_{V,h}^{(i)}\}_{h=1}^{H})$ and $\mathsf{ffn}(\cdot, W_{A,1}^{(i)}, W_{A,2}^{(i)})$ as $\mathsf{mha}^{(i)}(\cdot)$ and $\mathsf{ffn}^{(i)}(\cdot)$ in the following. The output logits of the transformer is

$$X^{(L+1)} = X^{(L)} W_{\mathsf{unemb}},$$

where $W_{\mathsf{unemb}} \in \mathbb{R}^{d \times d_{\mathsf{vocab}}}$ is the unembedding matrix. We would like to adopt the last row of $X^{(L+1)}$ to decode the next token. The parameters of the whole transformer is denoted as $\theta = \{W_{Q,h}^{(i)}, W_{K,h}^{(i)}, W_{V,h}^{(i)}\}_{i,h=1}^{L,H} \cup \{W_{A,1}^{(i)}, W_{A,2}^{(i)}\}_{i=1}^{L} \cup \{W_{\mathsf{unemb}}\}$. Then the whole transformer is denoted as

$$X^{(L+1)} = \mathsf{transformer}(X, \theta).$$

In our method, we will apply a mask on the MHA in some layers, where we only remain the first and last several tokens. This can be described by defined the masked indexes set $\mathcal{M}_i \subseteq [i]$ for $i$-row for $i \in [N]$. The corresponding mask $M_{\mathsf{lazy}}$ can be defined as

$$[M_{\mathsf{lazy}}]_{i,j} = \begin{cases} 0 & \text{if } j \notin \mathcal{M}_i \\ -\infty & \text{otherwise.} \end{cases}$$

For example, in our experiments, we set $\mathcal{M}_i$ as the first 4 and the last 1020 tokens. Then we denote the corresponding MHA as

$$\widetilde{\mathsf{mha}}\Big(X, \{W_{Q,h}, W_{K,h}, W_{V,h}\}_{h=1}^{H}\Big) = \sum_{h=1}^{H} \mathsf{softmax}(X W_{Q,h} W_{K,h}^{\top} X^{\top} + M_{\mathsf{lazy}}) X W_{V,h}.$$

The $\widetilde{\mathsf{mha}}$ module at $i$-th layer will be denoted as $\widetilde{\mathsf{mha}}^{(i)}$. The FF module will remain the same in the our method. We denote the set of indexes of the layers that apply this mask as $\mathcal{I}$. Then our method can be expressed as

$$\tilde{Y}^{(i)} = \tilde{X}^{(i-1)} + \mathbb{I}\{i \notin \mathcal{I}\} \cdot \mathsf{mha}^{(i)}\big(\mathsf{LN}(\tilde{X}^{(i-1)})\big) + \mathbb{I}\{i \in \mathcal{I}\} \cdot \widetilde{\mathsf{mha}}^{(i)}\big(\mathsf{LN}(\tilde{X}^{(i-1)})\big),$$

where we denote all the hidden states with our method applied as $\tilde{X}$ and $\tilde{Y}$, and $\mathbb{I}\{\cdot\}$ is the indicator function. The output of the whole network is denoted

$$\tilde{X}^{(L+1)} = \widetilde{\mathsf{transformer}}(X, \theta, \mathcal{I}).$$

To derive the theoretical analysis of the error, we need to delineate the norm of the transformer parameters. In fact, all the transformers in the real life have bounded parameters due to the calculation and storage requirements of the computer.

**Assumption F.2.** The Frobenius norms of all the parameters of the transformer is upper bounded by $B > 0$, i.e., $\|W_{Q,h}^{(i)}\|_{\mathbf{F}} \leq B$, $\|W_{K,h}^{(i)}\|_{\mathbf{F}}, \leq B$, $\|W_{V,h}^{(i)}\|_{\mathbf{F}} \leq B$, $\|W_{A,2}^{(i)}\|_{\mathbf{F}} \leq B$, $\|W_{A,1}^{(i)}\|_{\mathbf{F}} \leq B$, $\|W_{\mathsf{unemb}}\|_{\mathbf{F}} \leq B$ for $h \in [H]$ and $i \in [L]$.

To state our main result, we define the maximal *sum* of the original attention scores of the discarded tokens at layer $l \in \mathcal{I}$ as $s_l$, which is formally defined as

$$
\begin{aligned}
s_l &= \max_{i \in [N]} \frac{1}{H} \sum_{h=1}^{H} \left(1 - \sum_{j \notin \mathcal{M}_i} \frac{\exp\left(\left[\mathsf{LN}(X^{(l-1)})\right]_{i,:} W_{Q,h}^{(i)} W_{K,h}^{(i),\top} \left[\mathsf{LN}(X^{(l-1)})^{\top}\right]_{:,j}\right)}{\sum_{k=1}^{i} \exp\left(\left[\mathsf{LN}(X^{(l-1)})\right]_{i,:} W_{Q,h}^{(i)} W_{K,h}^{(i),\top} \left[\mathsf{LN}(X^{(l-1)})^{\top}\right]_{:,k}\right)}\right) \\
&= \max_{i \in [N]} \frac{1}{H} \sum_{h=1}^{H} \sum_{j \in \mathcal{M}_i} \frac{\exp\left(\left[\mathsf{LN}(X^{(l-1)})\right]_{i,:} W_{Q,h}^{(i)} W_{K,h}^{(i),\top} \left[\mathsf{LN}(X^{(l-1)})^{\top}\right]_{:,j}\right)}{\sum_{k=1}^{i} \exp\left(\left[\mathsf{LN}(X^{(l-1)})\right]_{i,:} W_{Q,h}^{(i)} W_{K,h}^{(i),\top} \left[\mathsf{LN}(X^{(l-1)})^{\top}\right]_{:,k}\right)}.
\end{aligned}
$$

Then the main result is as follows.

**Theorem F.3.** *We define the difference of the hidden states of our method and the original transformer at layer $i \in [L]$ as $e_X^{(i)} = \|X^{(i)} - \tilde{X}^{(i)}\|_{2,\infty}$. Under Assumptions F.1 and F.2, this error involves as*

$$e_X^{(i)} \leq e_X^{(i-1)} + \big(HB + L_{\mathsf{lip}}B^2 + 4HB^3\big) \min\left\{2, \left[1 + HB(1 + 4B^2)\right]e_X^{(i-1)}\right\} + 2H(B + L_{\mathsf{lip}}B^3)\mathbb{I}\{i \in \mathcal{I}\}s_i. \quad (4)$$

*The error between the logits generated by our method and the original transformer can be upper-bounded as*

$$\left\|\widetilde{\mathsf{transformer}}(X, \theta, \mathcal{I}) - \mathsf{transformer}(X, \theta)\right\|_{2,\infty} \leq 2LB^2\big(H + L_{\mathsf{lip}}B + 4HB^2\big) + 2HB^2(1 + L_{\mathsf{lip}}B^2)\sum_{i \in \mathcal{I}} s_i. \quad (5)$$

We note that the error recursive expression consists of three terms. The first term represents the error from the previous layer. The second term represents the error from the previous layer amplified by the current layer. Thanks to the layer normalization, this term will be truncated by 2. The last term represents the newly introduced error if we shorten KV cache at the current layer. By relaxing this recursive formula, we derive the upper bound of the error between logits of our method and the original transformer. This shows that the error is upper bounded by the sum of the attention scores of the removed KV pairs up to an additive constant.

*Proof of Theorem F.3.* We derive the error analysis of our analysis in three steps.

- The error decomposition of the whole network.

- Bound each term in the error decomposition.

- Conclude the proof.

**Step 1: The error decomposition of the whole network.**

We derive the error decomposition of the whole network in a recursive manner. In fact, for the $i$-th layer, we have that

$$\|\tilde{X}^{(i)} - X^{(i)}\|_{2,\infty} \leq \|\tilde{Y}^{(i)} - Y^{(i)}\|_{2,\infty} + \left\|\mathsf{ffn}^{(i)}\big(\mathsf{LN}(\tilde{Y}^{(i)})\big) - \mathsf{ffn}^{(i)}\big(\mathsf{LN}(Y^{(i)})\big)\right\|_{2,\infty}$$

$$\|\tilde{Y}^{(i)} - Y^{(i)}\|_{2,\infty} \leq \|\tilde{X}^{(i-1)} - X^{(i-1)}\|_{2,\infty} \tag{6}$$

$$+ \left\|\mathbb{I}\{i \notin \mathcal{I}\} \cdot \mathsf{mha}^{(i)}\big(\mathsf{LN}(\tilde{X}^{(i-1)})\big) + \mathbb{I}\{i \in \mathcal{I}\} \cdot \widetilde{\mathsf{mha}}^{(i)}\big(\mathsf{LN}(\tilde{X}^{(i-1)})\big) - \mathsf{mha}^{(i)}\big(\mathsf{LN}(X^{(i-1)})\big)\right\|_{2,\infty}, \tag{7}$$

where the inequalities follow from the triangle inequality. In addition, we have that

$$\left\|\widetilde{\mathsf{transformer}}(X, \theta, \mathcal{I}) - \mathsf{transformer}(X, \theta)\right\|_{2,\infty} \leq \|W_{\mathrm{unemb}}\|_{\mathbf{F}} \cdot \|X^{(L)} - \tilde{X}^{(L)}\|_{2,\infty}, \tag{8}$$

where the inequality results from Lemma G.2.

**Step 2: Bound each term in the error decomposition**

We will bound each term in the right-hand side of Eqn. equation 6 and equation 7. For the term related to the FF module, we have that

$$\left\|\mathsf{ffn}^{(i)}\big(\mathsf{LN}(\tilde{Y}^{(i)})\big) - \mathsf{ffn}^{(i)}\big(\mathsf{LN}(Y^{(i)})\big)\right\|_{2,\infty}$$

$$\leq L_{\mathsf{lip}} \cdot \|W_{A,2}^{(i)}\|_{\mathbf{F}} \cdot \|W_{A,1}^{(i)}\|_{\mathbf{F}} \cdot \left\|\mathsf{LN}(\tilde{Y}^{(i)}) - \mathsf{LN}(Y^{(i)})\right\|_{2,\infty}$$

$$\leq L_{\mathsf{lip}} \cdot \|W_{A,2}^{(i)}\|_{\mathbf{F}} \cdot \|W_{A,1}^{(i)}\|_{\mathbf{F}} \cdot \min\big\{2, \|\tilde{Y}^{(i)} - Y^{(i)}\|_{2,\infty}\big\}$$

$$\leq L_{\mathsf{lip}} \cdot B^2 \cdot \min\big\{2, \|\tilde{Y}^{(i)} - Y^{(i)}\|_{2,\infty}\big\}, \tag{9}$$

where the first inequality results from Lemma G.2, the second inequality results from the definition of $\ln(\cdot)$, and the last inequality results from Assumption F.2. For the term related to MHA module in the right-hand side of Eqn. equation 7, we have that

$$\left\|\mathbb{I}\{i \notin \mathcal{I}\} \cdot \mathsf{mha}^{(i)}\big(\mathsf{LN}(\tilde{X}^{(i-1)})\big) + \mathbb{I}\{i \in \mathcal{I}\} \cdot \widetilde{\mathsf{mha}}^{(i)}\big(\mathsf{LN}(\tilde{X}^{(i-1)})\big) - \mathsf{mha}^{(i)}\big(\mathsf{LN}(X^{(i-1)})\big)\right\|_{2,\infty}$$

$$= \mathbb{I}\{i \notin \mathcal{I}\} \cdot \left\|\mathsf{mha}^{(i)}\big(\mathsf{LN}(\tilde{X}^{(i-1)})\big) - \mathsf{mha}^{(i)}\big(\mathsf{LN}(X^{(i-1)})\big)\right\|_{2,\infty}$$

$$+ \mathbb{I}\{i \in \mathcal{I}\} \cdot \left\|\widetilde{\mathsf{mha}}^{(i)}\big(\mathsf{LN}(\tilde{X}^{(i-1)})\big) - \mathsf{mha}^{(i)}\big(\mathsf{LN}(X^{(i-1)})\big)\right\|_{2,\infty}$$

$$\leq \mathbb{I}\{i \notin \mathcal{I}\} \cdot \left\|\mathsf{mha}^{(i)}\big(\mathsf{LN}(\tilde{X}^{(i-1)})\big) - \mathsf{mha}^{(i)}\big(\mathsf{LN}(X^{(i-1)})\big)\right\|_{2,\infty}$$

$$+ \mathbb{I}\{i \in \mathcal{I}\} \cdot \left(\left\|\widetilde{\mathsf{mha}}^{(i)}\big(\mathsf{LN}(\tilde{X}^{(i-1)})\big) - \mathsf{mha}^{(i)}\big(\mathsf{LN}(\tilde{X}^{(i-1)})\big)\right\|_{2,\infty}\right.$$

$$\left. + \left\|\mathsf{mha}^{(i)}\big(\mathsf{LN}(\tilde{X}^{(i-1)})\big) - \mathsf{mha}^{(i)}\big(\mathsf{LN}(X^{(i-1)})\big)\right\|_{2,\infty}\right)$$

$$\leq H \cdot B\big(1 + 4B^2\big)\left\|\mathsf{LN}(X^{(i-1)}) - \mathsf{LN}(\tilde{X}^{(i-1)})\right\|_{2,\infty} + \mathbb{I}\{i \in \mathcal{I}\} \cdot 2BH \cdot s_i$$

$$\leq H \cdot B\big(1 + 4B^2\big)\min\big\{2, \|X^{(i-1)} - \tilde{X}^{(i-1)}\|_{2,\infty}\big\} + \mathbb{I}\{i \in \mathcal{I}\} \cdot 2BH \cdot s_i, \tag{10}$$

where the first inequality results from the triangle inequality, the second inequality results from Lemma G.4. Define the error $e_X^{(i)} = \|X^{(i)} - \tilde{X}^{(i)}\|_{2,\infty}$ with $e_X^{(0)} = 0$. Combining Eqn. equation 6, equation 7, equation 9, and equation 10, we have that

$$e_X^{(i)} \leq e_X^{(i-1)} + HB(1 + 4B^2)\min\{2, e_X^{(i-1)}\} + \mathbb{I}\{i \in \mathcal{I}\}2BHs_i$$

$$+ L_{\mathsf{lip}}B^2\min\big\{2, e_X^{(i-1)} + HB(1 + 4B^2)\min\{2, e_X^{(i-1)}\}\big\} + \mathbb{I}\{i \in \mathcal{I}\}2BHs_i\}. \tag{11}$$

**Step 3: Conclude the proof.**

We derive the recursive expression of the hidden state error by relaxing the right-hand side of Eqn. equation 11 as follows.

$$
\begin{aligned}
e_X^{(i)} &\leq e_X^{(i-1)} + HB(1 + 4B^2)\min\{2, e_X^{(i-1)}\} + \mathbb{I}\{i \in \mathcal{I}\}2BHs_i \\
&\quad + L_{\text{lip}}B^2 \min\left\{2, \left[1 + HB(1 + 4B^2)\right]e_X^{(i-1)}\right\} + \mathbb{I}\{i \in \mathcal{I}\}2L_{\text{lip}}B^3Hs_i \\
&\leq e_X^{(i-1)} + \left(HB + L_{\text{lip}}B^2 + 4HB^3\right)\min\left\{2, \left[1 + HB(1 + 4B^2)\right]e_X^{(i-1)}\right\} + 2H(B + L_{\text{lip}}B^3)\mathbb{I}\{i \in \mathcal{I}\}s_i.
\end{aligned}
$$

This proves the recursive formula. By summing this inequality from $i = 1$ to $i = L$, we have that

$$
e_X^{(L)} \leq 2L\left(HB + L_{\text{lip}}B^2 + 4HB^3\right) + 2(B + L_{\text{lip}}B^3)\sum_{i \in \mathcal{I}} s_i. \tag{12}
$$

Combining Eqn. equation 8 and equation 12, we have that

$$
\left\|\widetilde{\text{transformer}}(X, \theta, \mathcal{I}) - \text{transformer}(X, \theta)\right\|_{2,\infty} \leq 2LB^2\left(H + L_{\text{lip}}B + 4HB^2\right) + 2B^2H(1 + L_{\text{lip}}B^2)\sum_{i \in \mathcal{I}} s_i.
$$

Thus, we conclude the proof of Theorem F.3.

$\square$

# G   Supporting Lemmas

**Lemma G.1** (Corollary A.7 in Edelman et al. (2022) ). *For any $x, y \in \mathbb{R}^d$, we have*

$$
\|\text{softmax}(x) - \text{softmax}(y)\|_1 \leq 2\|x - y\|_\infty.
$$

**Lemma G.2** (Lemma 17 in Zhang et al. (2022) ). *Given any two conjugate numbers $u, v \in [1, \infty]$, i.e., $\frac{1}{u} + \frac{1}{v} = 1$, and $1 \leq p \leq \infty$, for any $A \in \mathbb{R}^{r \times c}$ and $x \in \mathbb{R}^c$, we have*

$$
\|Ax\|_p \leq \|A^\top\|_{p,u}\|x\|_v \quad \text{and} \quad \|Ax\|_p \leq \|A\|_{u,p}\|x\|_v.
$$

**Lemma G.3** (Lemma I.8 in Zhang et al. (2023)). *For any $X, \tilde{X} \in \mathbb{R}^{N \times d}$, and any $W_{Q,h}, W_{K,h} \in \mathbb{R}^{d \times d_h}, W_{V,h} \in \mathbb{R}^{d \times d}$ for $h \in [H]$ , if $\|X\|_{2,\infty}, \|\tilde{X}\|_{2,\infty} \leq B_X$, $\|W_{Q,h}\|_{\mathbf{F}} \leq B_Q$, $\|W_{K,h}\|_{\mathbf{F}}, \leq B_K$, $\|W_{V,h}\|_{\mathbf{F}} \leq B_V$ for $h \in [H]$, then we have*

$$
\begin{aligned}
&\left\|\text{mha}\left(X, \{W_{Q,h}, W_{K,h}, W_{V,h}\}_{h=1}^H\right) - \text{mha}\left(\tilde{X}, \{W_{Q,h}, W_{K,h}, W_{V,h}\}_{h=1}^H\right)\right\|_{2,\infty} \\
&\leq H \cdot B_V\left(1 + 4B_X^2 \cdot B_Q B_K\right)\|X - \tilde{X}\|_{2,\infty}.
\end{aligned}
$$

**Lemma G.4.** *For a query vector $q \in \mathbb{R}^d$, and two sets of key-value pairs $K_1 \in \mathbb{R}^{N_1 \times d}$, $K_2 \in \mathbb{R}^{N_2 \times d}$, $V_1 \in \mathbb{R}^{N_1 \times d}$, and $V_2 \in \mathbb{R}^{N_2 \times d}$, We define attention scores $\text{softmax}(q^\top[K_1, K_2]^\top)$ and $\text{softmax}(q^\top K_1^\top)$ as*

$$
\text{softmax}(q^\top[K_1, K_2]^\top) = [s_1^\top, s_2^\top], \text{ and } \text{softmax}(q^\top K_1^\top) = \tilde{s}_1^\top.
$$

*Then we have that*

$$
\left\|\text{softmax}(q^\top K_1^\top)V_1 - \text{softmax}(q^\top[K_1, K_2]^\top)[V_1^\top, V_2^\top]^\top\right\|_2 \leq 2\|s_2\|_1 \cdot \max\{\|V_1\|_{2,\infty}, \|V_2\|_{2,\infty}\}.
$$

*Proof of Lemma G.4.* In fact, we have that

$$
\text{softmax}(q^\top[K_1, K_2]^\top)[V_1^\top, V_2^\top]^\top = s_1^\top V_1 + s_2^\top V_2, \text{ and } \text{softmax}(q^\top K_1^\top)V_1 = \tilde{s}_1^\top V_1.
$$

Further, the difference between $s_1$ and $\tilde{s}_1$ can be upper bounded as

$$\|s_1 - \tilde{s}_1\|_1$$
$$= \sum_{i=1}^{N_1} \left| \frac{\exp(q^\top [K_1]_{i,:})}{\sum_{j=1}^{N_1} \exp(q^\top [K_1]_{j,:}) + \sum_{l=1}^{N_2} \exp(q^\top [K_2]_{l,:})} - \frac{\exp(q^\top [K_1]_{i,:})}{\sum_{j=1}^{N_1} \exp(q^\top [K_1]_{j,:})} \right|$$
$$= \sum_{i=1}^{N_1} \frac{\exp(q^\top [K_1]_{i,:}) \sum_{l=1}^{N_2} \exp(q^\top [K_2]_{l,:})}{\left( \sum_{j=1}^{N_1} \exp(q^\top [K_1]_{j,:}) + \sum_{l=1}^{N_2} \exp(q^\top [K_2]_{l,:}) \right) \sum_{j=1}^{N_1} \exp(q^\top [K_1]_{j,:})}$$
$$= \|s_2\|_1,$$

where the first equality results from the definition of $\mathsf{softmax}(\cdot)$, and the last equality results from the definition of $s_2$. Then we have that

$$\left\| \mathsf{softmax}(q^\top K_1^\top) V_1 - \mathsf{softmax}(q^\top [K_1, K_2]^\top)[V_1^\top, V_2^\top]^\top \right\|_2$$
$$= \left\| s_1^\top V_1 + s_2^\top V_2 - \tilde{s}_1^\top V_1 \right\|_2$$
$$\leq \|s_1 - \tilde{s}_1\|_1 \cdot \|V_1\|_{2,\infty} + \|s_2\|_1 \cdot \|V_2\|_{2,\infty}$$
$$\leq 2\|s_2\|_1 \cdot \max\{\|V_1\|_{2,\infty}, \|V_2\|_{2,\infty}\}.$$

Thus, we conclude the proof of Lemma G.4.

$\square$

