# OpenReview forum: "LightTransfer: Your Long-Context LLM is Secretly a Hybrid Model with Effortless Adaptation"
_TMLR — Accepted by TMLR_

### Review · Reviewer_qcXM · 2025-07-13

**Summary Of Contributions:**

The paper analyzes the behavior and weight structure of transformer on relatively long input sequences. The paper discovers that certain layers focus almost only on the initial and most recent tokens, which are called lazy layers in the paper. Inspired by this discovery, the paper proposes LightTransfer-Test, a runtime technique that converts lazy layers from full self-attention to streaming attention, significantly reducing key-value cache usage and improve inference throughput without additional training. Moreover, to handle reasoning tasks sensitive to attention changes, the paper introduces LightTransfer-Train, which fine-tunes only the lazy layers on a few thousand examples to restore performance. The paper then presents a bound on the logit-level approximation error introduced by these methods. Finally, the paper conducts experiment on multiple open-source models (LLaMA, Mistral) and benchmarks (long-context understanding, challenging math tests). The methods demonstrate up to 2.17× speedups, roughly 50% cache reduction, under 1.5% accuracy degradation, and even improved performance on math datasets.

**Audience:**

Yes

**Broader Impact Concerns:**

I do not foresee any ethical issues in this paper.

**Claims And Evidence:**

Yes

**Requested Changes:**

The paper should validate the lazy-ratio across diverse domains and measure its computation overhead and extend experiments to truly web-scale contexts to ensure generality. To strengthen the work, the paper can include an empirical study of the error bound’s tightness, analyze how accuracy degrades as more layers are converted to guide threshold selection, and quantify domain-specific fine-tuning requirements for LightTransfer-Train.

**Strengths And Weaknesses:**

Please find the strengths below:
- The paper defines a clear metric to pinpoint layers that ignore most of the context, revealing a previously underexplored structural property of long-context Transformers.
- The paper introduces LightTransfer-Test and LightTransfer-Train, which improve inference throughput without harming performance, even on reasoning tasks.
- The paper provides a logit-level approximation guarantee linking the discarded attention mass $s_i$ directly to the worst-case output deviation.

Please find the weaknesses below:
- The lazy-ratio metric was only evaluated on a limited set of datasets and tasks, so its behavior and computation overhead across different domains remain unclear.
- Experiments use contexts up to tens of thousands of tokens only; multi-document or web-scale inputs are untested and may yield different results.
- The theoretical error bound lacks any experimental or ablation study to validate its tightness or practical relevance.
- There is no systematic investigation of how many layers can be converted before accuracy drops sharply, leaving users without clear guidance on setting conversion thresholds.

---

> ### Author Response · Authors · 2025-08-01
>
> Thank you for your valuable feedback and questions. Below, we respond to the comments in Weaknesses (**W**) and Requested Changes (**RC**).
>
> ---
> ### W1. The lazy-ratio metric was only evaluated on a limited set of datasets and tasks, so its behavior and computation overhead across different domains remain unclear.
>
> Our current evaluation already covers a diverse set of 16 tasks spanning six distinct domains (Single-Doc QA, Multi-Doc QA, Summarization, Few-shot learning, Synthetic tasks, and Code completion) from LongBench, as well as the NIAH task from Ruler and the LongCoT generation task, thus providing a comprehensive evaluation across various domains and settings.
>
>
> ---
>
> ### W2. Experiments use contexts up to tens of thousands of tokens only; multi-document or web-scale inputs are untested and may yield different results.
>
> To further demonstrate the effectiveness of our method on extremely long contexts, we additionally conducted experiments using LLaMA3-8B-Instruct-Gradient-1048K on Ruler benchmark tasks with input lengths of 128K tokens:
> |Method|NIAH_single|NIAH_multikey|
> |-|-|-|
> |Vanilla|100.0|99.0|
> |MiniCache|14.6|14.6|
> |SqeezeAttn|98.0|69.0|
> |LightTransfer (ours)|99.6|96.2|
>
> As shown in the table, our LightTransfer also performs well with 128K inputs, outperforming other baselines.
>
> Moreover, in Table 2, we have already included comparisons between LightTransfer and baselines on three multi-document QA tasks (HotpotQA, Musique, DuReader) and three datasets containing web-scale inputs (TREC, TriviaQA, MF-en), further confirming our method’s robustness in handling multi-document and web-scale inputs. For example, DuReader contains questions and documents collected from Baidu Search and Baidu Zhidao, which are practical and realistic web data sources. The answers are manually annotated, making DuReader a challenging and representative benchmark for real-world QA applications.
>
>
>
> ---
>
> ### W3. The theoretical error bound lacks any experimental or ablation study to validate its tightness or practical relevance.
>
> In the revised manuscript (Section 5.3 and Appendix B.4), we have added experiments verifying the bound from Theorem 5.1. Specifically, we measure KL divergence between the standard and LightTransfer distributions across seven representative LongBench datasets, obtaining near-zero KL values (~$10^{-5}$ or lower). These results empirically confirm the bound's tightness and demonstrate its practical value: a tight bound assures that replacing standard attention with our lightweight module causes minimal distributional drift. Additionally, the results also indicate that the more layers we reduce, the larger the KL divergence becomes, consistent with our theoretical derivation of the bound.
>
>
> ---
>
> ### W4. There is no systematic investigation of how many layers can be converted before accuracy drops sharply, leaving users without clear guidance on setting conversion thresholds.
>
> As illustrated in Figure 6 and Section 6.3, we evaluated the model performance by systematically varying the fraction of layers using original attention from 0.25 to 0.5, and up to 0.75, for both LLaMA3-8B-Instruct and LLaMA3-70B-Instruct on the LongBench benchmark. This analysis provides guidance for users on choosing appropriate conversion thresholds.
>
>
> ---
>
> ### RC1.The paper should validate the lazy-ratio across diverse domains and measure its computation overhead and extend experiments to truly web-scale contexts to ensure generality.
>
> As detailed in our response to W1, we have provided evaluation across diverse domains and web-scale contexts. Regarding computational overhead, Section 5.1 demonstrates that the throughput reduction introduced by LightTransfer-Test is minimal, only ranging from 0.0058 to 0.0014 relative to the baseline of 1 across sequence lengths from 4K to 32K during prefilling. Furthermore, LightTransfer requires only 3000 data samples for SFT, highlighting its efficiency and practicality
>
>
>
> ---
>
> ### RC2. To strengthen the work, the paper can include an empirical study of the error bound’s tightness, analyze how accuracy degrades as more layers are converted to guide threshold selection, and quantify domain-specific fine-tuning requirements for LightTransfer-Train.
>
> Please see our responses to W3 and W4 for analyses regarding the error bound and accuracy degradation as more layers are converted.
>
> Regarding domain-specific fine-tuning requirements, the publicly available LongCoT corpus, lacks fine‑grained domain annotations beyond broad mathematics‑centric tasks. This prevents us from conducting a rigorous domain‑wise ablation at present. Nevertheless, we emphasize that upgrading from LightTransfer‑Test to LightTransfer‑Train requires only 3000 SFT samples, demonstrating the method’s low data footprint even when domain labels are scarce.

---

### Review · Reviewer_tEVS · 2025-07-18

**Summary Of Contributions:**

The paper introduces LightTransfer, a lightweight framework that transforms pretrained transformer LLMs into hybrid models by:
- Identifying “lazy” layers—those that predominantly attend only to the first few or most recent tokens—and replacing their full self‐attention with memory‐efficient streaming attention.
- Zero‐shot adaptation for long‐context understanding tasks (no additional training required) and minimal fine‐tuning (≈5 K examples) for advanced long‐reasoning generation tasks.
- Extensive empirical evaluation across multiple LLMs (LLaMA 2 7B‐chat, Mistral 7B‐Instruct, LLaMA 3 8B/70B) on benchmarks including LongBench, Needle‐In‐A‐Haystack, and AIME24. Results show up to 2.17× token‐generation throughput improvement with < 1.5 % performance drop on LongBench and 53.3 % accuracy on AIME24—matching or exceeding baselines while replacing half the layers .
- Ablation studies demonstrating the trade‐off between the fraction of retained full‐attention layers and performance, and superiority over other inter‐layer KV‐cache reduction methods .
- Theoretical grounding, optimizing an explicit error upper bound (Theorem 5.1) to guide layer replacement decisions, and analysis of attention patterns in long‐context inference

**Audience:**

Yes

**Claims And Evidence:**

No

**Requested Changes:**

- Clarify experimental setup, particularly around sink tokens
- Extend the theoretical analysis further
- Discuss limitations and address the ambiguities in the plots and text

**Strengths And Weaknesses:**

# Strengths
- Efficiency without retraining
LightTransfer delivers substantial memory savings and up to a 2.17× increase in token‐generation throughput with no or minimal additional training, making it straightforward to drop into existing pretrained transformers.
- Strong empirical performance
Across multiple models (LLaMA 2, Mistral, LLaMA 3) and long-context benchmarks, it maintains accuracy within 1.5 % of full self-attention while more than doubling throughput on sequences up to 32 K tokens.
- Broad applicability
Validated on diverse architectures (e.g. LLaMA, Mistral, QwQ-STILL) and tasks (zero-shot understanding, fine-tuned reasoning), demonstrating robustness across model sizes and use cases.
- Insightful analysis
Clear visualizations of layer-wise attention patterns expose which layers are “lazy,” and a principled method leverages these observations to decide where to substitute streaming attention.
- Solid theoretical underpinning
An explicit error bound (Theorem 5.1) guides layer replacement decisions and distinguishes LightTransfer from purely heuristic approaches.

# Weaknesses

## Major
- Ambiguity around “streaming attention” setup

The original streaming-attention work uses learnable sink tokens and requires retraining. Here, however, the paper treats the first few prompt tokens as “sink” tokens without retraining. The authors should (a) clearly describe their implementation (are sink tokens learned or fixed?), and (b) compare performance against a fully-trained streaming-attention baseline to quantify any trade-offs.
- Unclear use of the theoretical error bound

Section 5.3 derives an upper bound on the error introduced by removing KV caches, but it isn’t clear how to translate that bound into practice. How can one use Eqn 1 to predict model quality drop for a given layer replacement ratio? Concrete guidance or examples would elevate the theorem from an observation to an actionable design tool.

## Minor
-  Overhead measurement in Section 5.1

The statement “throughput reduction of 0.0058 to 0.0014 relative to a baseline of 1” is opaque. Does this correspond to a 0.14 – 0.58 % overhead? Is it measured only during prefilling or across both prefilling and decoding?

- Figure 2 (left panel) clarity

The circular highlight around the border can be mistaken for the plot boundary. Consider adding a distinct rectangular border or subtle shading behind the highlight to separate it from the axes.

- Figure 2 (right panel) generation details

It’s unclear whether the heatmap reflects attention to all early and recent tokens or just one representative token. Please specify how many tokens were aggregated, whether the color scale is an average or maximum, and how the data points were computed (e.g. per-token mean over a batch).

- No limitation discussed

---

> ### Author Response · Authors · 2025-08-01
>
> Thank you for your valuable feedback and questions. Below, we respond to the comments in Weaknesses (**W**) and Requested Changes (**RC**).
>
> ---
> ### W1 & RC1. Ambiguity around “streaming attention” setup.
> >The original streaming-attention work uses learnable sink tokens and requires retraining. Here, however, the paper treats the first few prompt tokens as “sink” tokens without retraining. The authors should (a) clearly describe their implementation (are sink tokens learned or fixed?), and (b) compare performance against a fully-trained streaming-attention baseline to quantify any trade-offs.
>
> We provide two variants:
>
> *LightTransfer‑Test.* This training‑free variant reuses the sink‑token parameters (i.e., sink token corresponded embeddings) that were already **learned during the original pre‑training of the backbone model**. In other words, the first K input tokens are treated as sink tokens, and their embeddings/weights remain exactly as in the released checkpoint, mirroring the setup of the original Streaming‑Attention paper (Section 4.3).
>
> *LightTransfer‑Train.* During SFT, all parameters, including **the sink‑token embeddings**, are updated. We do not freeze or randomly re‑initialize any sink‑token parameters. Consequently, the sink tokens continue to be learnable and adapt to downstream data.
>
>
>
> ---
>
> ### W2 & RC2. Unclear use of the theoretical error bound.
> >Section 5.3 derives an upper bound on the error introduced by removing KV caches, but it isn’t clear how to translate that bound into practice. How can one use Eqn 1 to predict model quality drop for a given layer replacement ratio? Concrete guidance or examples would elevate the theorem from an observation to an actionable design tool.
>
> In the revised manuscript (Section 5.3 and Appendix B.4), we have added experiments verifying the bound from Theorem 5.1. Specifically, we measure KL divergence between the standard and LightTransfer distributions across seven representative LongBench datasets, obtaining near-zero KL values (~$10^{-5}$ or lower). These results empirically confirm the bound's tightness and demonstrate its practical value: a tight bound assures that replacing standard attention with our lightweight module causes minimal distributional drift. Additionally, the results also indicate that the more layers we reduce, the larger the KL divergence becomes, consistent with our theoretical derivation of the bound.
>
>
> ---
>
> ### W3 & RC3. Overhead measurement in Section 5.1
> > The statement “throughput reduction of 0.0058 to 0.0014 relative to a baseline of 1” is opaque. Does this correspond to a 0.14 – 0.58 % overhead? Is it measured only during prefilling or across both prefilling and decoding?
>
> Yes, the reported overhead corresponds to 0.14%–0.58% relative to the baseline throughput, and it was measured only during the prefilling stage. Decoding throughput remains unaffected.
>
>
> ---
>
> ### W4 & RC3. Figure 2 (left panel) clarity.
> >The circular highlight around the border can be mistaken for the plot boundary. Consider adding a distinct rectangular border or subtle shading behind the highlight to separate it from the axes.
>
> We have revised Figure 2 (left panel) by adding subtle shading behind the circular highlight. This improvement distinguishes the highlight from the plot boundary, enhancing visual clarity.
>
>
> ---
>
> ### W5 & RC3. Figure 2 (right panel) generation details
> >It’s unclear whether the heatmap reflects attention to all early and recent tokens or just one representative token. Please specify how many tokens were aggregated, whether the color scale is an average or maximum, and how the data points were computed (e.g. per-token mean over a batch).
>
> We have updated the caption of this figure. The heatmap aggregates attention weights from each query token (x-axis) to two token groups:(1) the first W_sink tokens, and (2) the most recent W_recent tokens. The color scale in each cell represents the sum of attention weights (not average or maximum) from the corresponding query token to these two token groups. The values shown are computed as the mean across all samples in a batch.
>
>
> ---
>
> ### W6 & RC3. No limitation discussed.
>
> We add a section in Appendix B.5 discussing these limitations explicitly.

---

### Review · Reviewer_pnfF · 2025-07-23

**Summary Of Contributions:**

The authors propose a method to identify the Transformers layers that pay great attention to unimportant tokens and the most recent ones, and replace the full attention mechanism of these layers with streaming attention. On top of four representative LLMs, the authors conduct comprehensive experiments on the long context understanding task and 1-like long reasoning generation tasks.

**Audience:**

Yes

**Claims And Evidence:**

No

**Requested Changes:**

See Weakness.

**Strengths And Weaknesses:**

I need to clarify first that I am not a full expert in this domain.

Strengths:
* The paper is well-organised and easy to read.
* The comprehensive experiments have been conducted.

Weaknesses:
* The idea of replacing the attention mechanism of some layers in Transformers with a more efficient one is not novel.
* The authors should also provide the experiments to verify the bound shown in Theorem 5.1 and should explain why this bound is useful.
* In *Effect of different layer replacement strategies* part, the authors should also compare other replacement strategies.

References:
* [Investigating Layer Importance in Large Language Models](https://aclanthology.org/2024.blackboxnlp-1.29/) (Zhang et al., BlackboxNLP 2024)
* [A Primer in BERTology: What we know about how BERT works](https://arxiv.org/abs/2002.12327) (Rogers et al. 2020)

---

> ### Author Response · Authors · 2025-08-01
>
> Thank you for your valuable feedback and questions. Below, we respond to the comments in Weaknesses (**W**) and Requested Changes (**RC**) .
>
> ---
>
> ### W1. The idea of replacing the attention mechanism of some layers in Transformers with a more efficient one is not novel.
>
> We respectfully clarify that the novelty of our method lies primarily in achieving *O(1)* complexity for lazy layer identification at test time and no degration on challenging long CoT generation tasks, while providing theoretical bounds on the error introduced relative to the original model. This ensures both computational efficiency and controlled accuracy loss, offering a principled and novel trade-off. Additionally, our method outperforms alternative layer-wise replacement methods such as MiniCache.
>
>
>
> ---
>
> ### W2. The authors should also provide the experiments to verify the bound shown in Theorem 5.1 and should explain why this bound is useful.
>
> In the revised manuscript (Section 5.3 and Appendix B.4), we have added experiments verifying the bound from Theorem 5.1. Specifically, we measure KL divergence between the standard and LightTransfer distributions across seven representative LongBench datasets, obtaining near-zero KL values (~$10^{-5}$ or lower). These results empirically confirm the bound's tightness and demonstrate its practical value: a tight bound assures that replacing standard attention with our lightweight module causes minimal distributional drift. Additionally, the results also indicate that the more layers we reduce, the larger the KL divergence becomes, consistent with our theoretical derivation of the bound.
>
>
> ---
>
> ### W3. In Effect of different layer replacement strategies part, the authors should also compare other replacement strategies.
>
> We appreciate the suggestion and have added comparison with the two alternatives to Section 6.3 and Appendix C.5. As shown in the updated results, both baselines are consistently inferior to our proposed strategy across all evaluation sets. These findings further demonstrate the advantages of our method.

---

### Review · Reviewer_9JNz · 2025-07-23

**Summary Of Contributions:**

The paper introduces a method to convert transformers to hybrid models (with different attention styles in different layers). The method LightTransfer has two methods, `test` for inference-time transformations and `train` for reasoning models, with a single-shot transformation. For the hybrid layer, they use streaming attention layers and show initial analysis supporting that streaming attention patterns emerge naturally for long context tasks. The `test` method, used for non-reasoning models, an on-the-fly transform is applied per sample. They define the lazy ratio as the metric to figure out which layers to set as lazy (ie, convert to hybrid). Using attention weight recomputation, they reduce the cost to compute the lazy ratios across the model. The `train` method, used for reasoning models, uses the data for prompt + completion to identify the layers in an offline manner, and then run SFT to recover accuracy. The method is shown to have good results on long context tasks (LongBench and NIAH) using the `test` method and math reasoning tasks (AIME / MATH-OAI / GSM8k).

**Audience:**

Yes

**Claims And Evidence:**

No

**Requested Changes:**

- Some of the wording around the experimentation section is unclear. It takes multiple reads to figure out which model was being used in the `train` method, and towards this, it will be better to be more upfront. [**strengthen the work**]
- Some more details on benchmarking will be useful. What inference engine setup, HW etc, since those details matter for inference performance. [**critical to secure recommendation**]

**Strengths And Weaknesses:**

**Strengths**
- The paper's method is simple to use and does not add too much overhead
- The eval benchmarks presented are relevant for scenarios where KV cache compression hurts the most (LongBench and NIAH)
- The training method is simple, it relies on simple SFT and also uses open datasets for recovering model quality.
- The paper provides sound theoretical analysis for their method and also the writeup in appendix B.2
- The reuse of attention recomputation is good as it minimizes overhead.

**Weaknesses**
- Some of the parts of the method was unclear. The authors say their method is calibration free for training, but then they pass the SFT data through the model to figure out lazy layers?
- Some of the parts of the paper are difficult to follow. Many tables such as Tables 3, 4, 7 are missing baseline numbers and Table 5 provide only speedup factors without providing details for actual numbers.
- There is minimal novelty in the method of replacing attention layers in transformers.

---

> ### Author Response · Authors · 2025-08-01
>
> Thank you for your valuable feedback and questions. Below, we respond to the comments in Weaknesses (**W**) and Requested Changes (**RC**) .
>
> ---
>
> ### W1. Some of the parts of the method was unclear.
> >The authors say their method is calibration free for training, but then they pass the SFT data through the model to figure out lazy layers?
>
> We clarify that LightTransfer has two distinct variants: LightTransfer-Test and LightTransfer-Train. The term *"calibration-free"* specifically applies to **LightTransfer-Test**, where lazy layers are dynamically determined on-the-fly by the input prompt with the \( O(1) \) identification algorithm (Table 1), without any reliance on held-out or SFT data.
>
>
> The usage of SFT data occurs **only** in the LightTransfer-Train variant, designed explicitly for training models on long-form CoT generation tasks.
>
> ---
>
> ### W2. Some of the parts of the paper are difficult to follow.
> > Many tables such as Tables 3, 4, 7 are missing baseline numbers and Table 5 provide only speedup factors without providing details for actual numbers.
>
> Tables 3, 4, and 7 focus exclusively on the LightTransfer-Train scenario, and thus use different baselines from LightTransfer-Test in Table 2.
>
> Regarding Table 5, we report only the speedup factors to facilitate clear comparisons. For completeness, we provide the actual Throughput numbers (tokens/s) below:
>
> | Method            | 4K   | 8K   | 32K |
> | ----------------- | ---- | ---- | --- |
> | Vanilla           | 2007 | 1045 | 260 |
> | SqueezeAttention  | 2067 | 1139 | 270 |
> | MiniCache         | 2529 | 1348 | 367 |
> | **LightTransfer** | 2908 | 1859 | 456 |
>
> ---
>
> ### W3. There is minimal novelty in the method of replacing attention layers in transformers.
>
> We respectfully clarify that the novelty of our method lies primarily in achieving *O(1)* complexity for lazy layer identification at test time and no degration on challenging long CoT generation tasks, while providing theoretical bounds on the error introduced relative to the original model. This ensures both computational efficiency and controlled accuracy loss, offering a principled and novel trade-off. Additionally, our method outperforms alternative layer-wise replacement methods such as MiniCache.
>
> ---
>
> ### RC1. Some of the wording around the experimentation section is unclear.
> >It takes multiple reads to figure out which model was being used in the train method, and towards this, it will be better to be more upfront.
>
> To address this concern, we have re‑written the settings paragraph of Section 6.2 to state upfront which model is fine‑tuned and why. This revision makes the training pipeline explicit and explains our choice of data and method.
>
> ---
>
> ### RC2. Some more details on benchmarking will be useful.
> >What inference engine setup, HW etc, since those details matter for inference performance.
>
> Following the reviewer’s suggestion, we have added more detailed experimental settings in Appendix A (page 16 in the revised paper). The updated information is as follows:
>
> *All the experiments are conducted using NVIDIA A100 using PyTorch and HuggingFace Transformers replaced with `flash_attention_with_kvcache` to accelerate. All model weights, activations, and KV caches use BF16 precision, with no quantization applied.*
>
> We will also make our code publicly available to facilitate reproduction.

---

> > ### Comment · Reviewer_9JNz · 2025-08-05
> > **some notes**
> >
> > I thank the authors for the reponses. After reading the additional information added to the paper, and the new results in the rebuttal, I have no further questions.

---

### Decision · Action_Editor_3DN9 · 2025-08-31

**Recommendation:** Accept with minor revision

**Additional Comments:**

The authors provided a thorough and effective rebuttal, including new experiments and significant revisions to the manuscript, which successfully addressed almost all reviewer concerns. The responses can be included in the revised version.

**Audience:**

Yes

**Audience Explanation:**

The paper tackles a highly relevant, unsolved problem in a popular and fast-moving area of LLMs. It provides a practical, empirically-validated solution that would interest both the applied and research-focused segments of the TMLR audience.

**Claims And Evidence:**

Yes

**Claims Explanation:**

The one point that remains is the question of novelty, raised by Reviewer 9JNz ("minimal novelty in the method of replacing attention layers"). The authors defended this by positioning the novelty in the O(1) identification algorithm and the principled theoretical backing, rather than the general idea of layer replacement. While Reviewer 9JNz still noted the novelty was "limited" in their final recommendation, they still found the work to be an "interesting empirical idea" and leaned towards acceptance. This suggests that the paper's empirical strength and practical value are sufficient to overcome concerns about fundamental novelty. The evidence presented now convincingly supports the paper's claims. The authors have been highly responsive and have materially improved the submission based on reviewer feedback. The initial weaknesses were significant, but they have been systematically and thoroughly addressed. The work presents a practical, well-validated method that offers significant efficiency gains with minimal performance trade-offs, backed by both strong empirical results (on contexts up to 128K) and now-validated theoretical grounding.

The authors provided a thorough and effective rebuttal, including new experiments and significant revisions to the manuscript, which successfully addressed almost all reviewer concerns. The responses can be included in the revised version.